

# Dissipation of the energy imparted by mid-latitude storms in the Southern Ocean

J. Jouanno,[1] X. Capet,[2] G. Madec,[2] G. Roullet[3], P. Klein[3], and S. Masson[2]

[1]LEGOS, IRD, Univ. Paul Sabatier, Observatoire Midi-Pyrénées, Toulouse, France

[2]LOCEAN, UPMC, Paris, France

[3]LPO, UBO, Brest, France

*Correspondence to*: J. Jouanno (jouanno@legos.obs-mip.fr)

**Abstract.** The aim of this study is to clarify the role of the Southern Ocean storms on interior mixing and meridional overturning circulation. A periodic and idealized numerical model has been designed to represent the key physical processes

of a zonal portion of the Southern Ocean located between 70°S and 40°S. It incorporates physical ingredients deemed essential for Southern Ocean functioning : rough topography, seasonally varying air-sea fluxes, and high-latitude storms with analytical form. Level and distribution of mixing attributable to high frequency winds are quantified and compared to those generated by eddy-topography interactions and dissipation of the balanced flow. Results suggest that 1) the synoptic atmospheric variability alone can generate the levels of mid-depth dissipation frequently observed in the Southern Ocean

($10^{-10}$ - $10^{-9}$ W kg$^{-1}$) and 2) the storms strengthen the overturning, primarily through enhanced mixing in the upper 300m, whereas deeper mixing has a minor effect. The sensitivity of the results to horizontal resolution (20, 5, 2 and 1 km), vertical resolution and numerical choices is evaluated. Challenging issues concerning how numerical models are able to represent interior mixing forced by high-frequency winds are exposed and discussed, particularly in the context of the overturning circulation. Overall, submesoscale-permitting ocean modelling exhibits important delicacies owing to a lack of convergence

of key components of its energetics even when reaching $\Delta x = 1$ km.

## 1 Introduction

Knowledge gaps pertaining to energy dissipation and mixing distribution in the ocean greatly limit our ability to apprehend its dynamical and biogeochemical functioning (globally or at smaller scale, e.g., regional) and its role in the climate system

evolution (Naveira-Garabato, 2012). For example, the meridional overturning circulation in low-resolution global coupled models is significantly altered by the parameterization for and intensity of vertical mixing (Jayne et al. 2009, Melet et al. 2013).

   A great deal of efforts is currently deployed to address the issue but the difficulties are immense: dissipation occurs intermittently, heterogeneously and in relation with a myriad of processes whose importance varies depending on the region,





depth range, season, proximity to bathymetric features... In this context, establishing an observational truth based on local estimates involves probing the ocean at cm scale (vertically) with horizontal and temporal resolution requirements that will need a long time to be met (e.g.,  MacKinnon et al. 2009 or DIMES program, Gille et al. 2012).

    In order to make progress other (non-exclusive) approaches are being followed. Well-constrained bulk mixing requirements for certain water masses can be exploited to infer mixing rates and, in some cases point to (or discard) specific processes (de Lavergne et al., submitted). Alternatively, in-depth investigations of dissipation and mixing associated with presumably important processes are carried out (with the subsequent parameterization of the effects in OGCMs being the ultimate objective, Jayne et al. 2009, Jochum et al. 2009). This study belongs to the latter thread. It is a numerical contribution to the investigation of dissipation and mixing due to atmospheric synoptic variability (mid-latitude storms) in the Southern Ocean.

    Synoptic or high-frequency winds inject important amounts of energy into the ocean that feed the near inertial wave (NIW) field. A large part of the near inertial energy (NIE) dissipates locally in the upper ocean, where it deepens the mixed-layer and potentially has an impact on the air-sea exchanges and global atmospheric circulation (Jochum et al. 2013). Nevertheless a substantial fraction of the NIE also spreads horizontally and vertically away from its source regions: beta dispersion propagates the energy toward lower latitudes (Anderson and Gill 1979), advection by the geostrophic circulation redistributes NIE laterally (Zhai et al. 2005), and the mesoscale eddy field favors the penetration of NIWs into the deep ocean by shortening their horizontal scales (Danioux et al. 2008, Zhai et al. 2005), or through the ¨inertial chimney¨ effect (Kunze 1985).

    Although the near inertial part of the internal wave spectrum is thought to contain most of the energy and vertical shear (Garrett 2001), large uncertainties remain on the amount of NIE available at depth for small-scale mixing and whether/where it is significant compared to other sources of mixing such as the breaking of internal waves generated by tides or the interaction of the mesoscale flow with rough topography (e.g. Nikurashin et al. 2013). The only present consensus is that NIE due to atmospheric forcing does not penetrate efficiently enough into the ocean interior to provide the mixing necessary to close the deep cells of the MOC (Furuichi et al. 2008, Ledwell et al. 2011), below 2000 m.

    On the other hand, the vertical flux of NIE at 800 m estimated by Alford et al (2012) at station Papa (in a place of the north Pacific not particularly affected by storm activity) may have significant implications on mixing of the interior water masses, depending on the (unknown) depth range where it dissipates. Our regional focus is the  Southern Ocean where intense storm activity forces NIW (Alford, 2003) that seem to have important consequences, at least above 1500 m depth. Elevated turbulence in the upper 1000-1500 meters north of Kerguelen plateau has been related to wind-forced downward-propagating near inertial waves (Waterman et al. 2013); the clear seasonal cycle of diapycnal mixing estimated from over 5000 ARGO profiles in regions of the Southern Ocean where topography is smooth points to the role of wind-input in the near-inertial range (and NIW penetration into the ocean interior; Wu et al, 2011).

    The aim of this study is to i) further clarify the mechanisms implicated in NIW penetration into the ocean interior, ii) more precisely quantify the resulting NIE dissipation intensity including its vertical distribution, and iii) better understand the



current (and future) OGCM limitations in representing NIE dissipation. (Findings on (ii) will be specific to the Southern Ocean while we expect those on (i) and (iii) to be more generic.) For that purpose we perform semi-idealized Southern Ocean simulations for a wide range of model parameters and different numerical schemes covering eddy present to submesoscale rich regimes.

5    Importantly, our highest resolution simulations adequately resolve the meso- and submeso-scale turbulent activity deemed essential in the leakage of NIE out of the surface layers, as found in Danioux et al (2011). In contrast to this and other studies (Danioux et al, 2008) the realism of the ocean forcing, mean state, and circulation makes it more directly applicable to the real ocean, provided that numerical robustness and convergence is reasonably achieved.

The paper is organized as follows. The model setup is presented in section 2 and the ocean dynamics and mean state that 10   are simulated without storms are described in section 3. Section 4 describes the spatial and temporal characteristics and consequences of the simplified NIW field generated by the passage of a single storm (spin down experiment). In section 5 quasi-equilibrated simulations are analyzed in terms of pathways through which the storm energy is deposited into the interior ocean and sensitivity of the mixing distribution to storm parameters and numerical choices. In section 6, we characterize the long term impact of the storms on the (large scale) MOC which turns out to be significant, mainly because 15   of their effect on and immediately below the ocean surface boundary layer. Section 7 provides some discussion and section 8 concludes.

## 2 Model

The numerical set up consists of a periodic channel configuration of 2000 km length (Lx, zonal direction) and 3000 km 20   width (Ly, meridional direction), that aims to represent a zonal portion of the Southern Ocean located between 70°S and 40°S (Fig. 1). It is inspired by the experiment described in Abernathey et al. (2011), which is mainly adiabatic in the interior. We add three ingredients to our reference experiment deemed essential to reach realistic levels of dissipation and whose consequence is to enhance dissipation and mixing in the model ocean interior.

*i) The bathymetry is random and rough.* Horizontal scales of the reference bathymetry range between 10 and 100 km and 25   depths vary between 3000 and 4000 meters. The bottom roughness,  computed as the variance of the bottom height H, is $3.10^4$ m², value which can be considered as intermediate between rough and smooth and which is representative of the roughness of a large portion of the Southern Ocean topography (see map of roughness in Wu et al. 2011). The inclusion of bottom topography aims to limit the ACC transport through bottom form stress (Rintoul et al. 2001) and to generate deep and mid-depth mixing through vertical shear. Our horizontal resolution $\geq$ 1km and the hydrostatic hypothesis used to derive the 30   model primitive equations corrupt the upward radiation of internal lee waves (Nikurashin et al., 2011) but deep flows impinging on bottom irregularities generate fine-scale shear which enhances dissipation and mixing close to the bottom (see section 5).



*ii) The surface and lateral forcing vary seasonally.* The objective is to reproduce a seasonally varying stratification and mixed-layer depth. These seasonal variations are known to be important in the formation process of mode waters and functioning of the overturning, since surface cooling triggers mixed-layer convection.

*iii) The wind forcing includes idealized Southern Ocean storms.* These high-frequency winds induce intense near-inertial energy and mixing into the ocean interior. From the analysis of scatterometer measurements, Patoux et al. (2009) provided general statistics of the spatial and temporal variability of the Southern Ocean mid-latitude cyclones for the period 1999-2006 : most of the cyclones occurred between 50ºS and 70ºS, have a radius between 400 and 800 km, and last between 12h and 5 days. Mesoscale cyclones lasting less than 4 days represent about 75% of all cyclone tracks (Yuan et al. 2009). The storm forcing design, detailed in Appendix A and adapting the methodology followed by Vincent et al. (2012), is based on these observations.

## 2.1 Configuration

The numerical code is the oceanic component of the Nucleus for European Modelling of the Ocean program (NEMO, Madec 2014). It solves the three dimensional primitive equations discretized on a C-grid and fixed vertical levels (z-coordinate). Horizontal resolution of the reference simulation is 2-km. There are 50 levels in the vertical (with 10 levels in the upper 100 meters and cells reaching a height of 175 m at the bottom). Sensitivity runs to both horizontal and vertical resolutions ($\Delta x$ between 1 and 20-km, $\Delta x$=2-km with 320 vertical levels) are an important part of this study. The model is run on $\beta$-plane with $f_0$=1.10$^{-4}$ s$^{-1}$ at the center of the domain and $\beta$= 1.10$^{-11}$ m$^{-1}$ s$^{-1}$. A 3rd order upstream biased scheme (UP3) is used for both tracer and momentum advection, with no explicit diffusion. The vertical diffusion coefficients are given by a Generic Length Scale (GLS) scheme with a *k-ε* turbulent closure (Reffray et al. 2015). Bottom friction is linear with a bottom drag coefficient of 1.5 10$^{-3}$ m s$^{-1}$. We use a linear equation of state only dependent on temperature with linear thermal expansion coefficient $\alpha$=2.10$^{-4}$ K$^{-1}$. The temporal integration is achieved by a modified Leap Frog Asselin Filter (Leclair and Madec 2009), with a coefficient of 0.1 and a time step of 150 seconds for the 2-km experiments. Sensitivity to these parameters and numerical choices are also performed.

Air-sea heat fluxes are built so as to represent the observed seasonal evolution of the zonally averaged sea surface temperature and mixed-layer depth in the Southern Ocean (Fig. 2a-b). The surface heat flux $Q_{net}$ is as follows : $Q_{net}=Q_{solar}+Q_{nonsolar}$ , with $Q_{solar}$ the shortwave heat flux and $Q_{nonsolar}$ the non solar heat flux accounting for the effect of longwave, latent, sensible heat fluxes, and a feedback term g ($T_{clim}$ -$T_{model}$). This feedback term depends on a sensitivity term g set to 30 W m$^{-2}$ K$^{-1}$ (Barnier et al. 1995) and on the difference between $T_{clim}$, a SST climatology which varies seasonally, and $T_{model}$ the model SST. The seasonal amplitude of $Q_{net}$ in the center of the domain is 200 W m$^{-2}$ (Fig. 2h), a value close to the observations (Fig. 2c). Over the northern 150 km of the domain, the temperature is relaxed toward an exponential temperature profile varying seasonally in the upper 150 meters. The response of the ocean to this forcing leads to a seasonal cycle of the surface temperature (Fig. 2f), and a deepening of the mixed-layer from 30 meters in summer to 150 meters in winter (Fig. 2g), in good agreement with zonally averaged observations of the Southern Ocean (Fig. 2a and 2b). It is worth mentioning that the



direct effect of a storm on the air-sea buoyancy flux (modulation of the radiative, latent and sensible heat fluxes) is not explicitly accounted for.

Two long reference experiments, one with storms and another without storms, with horizontal resolution 2-km have been run for 40 years. For these experiments, the model is started from a similar simulation without storms, equilibrated with a 200-year long spin-up at 5-km horizontal resolution. Unless otherwise stated, the last 10 years of the simulations are used for diagnostics, excluding the northern 150 km band where restoring is applied. Similar long term simulations with horizontal resolution 20-km and 5-km have also been performed in order to determine meridional overturning modifications with horizontal resolution (section 7).

An experiment with a single storm traveling eastward through the center of the basin over an equilibrated ocean has also been performed. Initial conditions are taken from the 2km horizontal resolution simulation (without storm) at day December 31 of year 30 from the 2-km reference experiment without storms. The storm is centered at the meridional position Ly/2 and has a maximum windstress of 1.5 N m$^{-2}$. The ocean spin-down response is analyzed for a period of 70 days (the storm is centered at days 5, starting at day 3 and ending at day 7).

In order to assess the sensitivity of interior mixing to numerics and storms characteristics, additional experiments have been run over shorter periods of 3 years, starting from year 30 of the 2-km reference experiment without storms. These experiments are summarized in Table 1 and will be analyzed in section 5. The last two years of these experiments are used for diagnostics. Although the model is not equilibrated after a period of 3 years, we have verified in section 5 that changes in terms of energy dissipation and mixing diagnosed over this short period are significant.

The averaged total wind work in the 2-km experiment with storms is 16.8 mW m$^{-2}$. This value is comparable to the 20 mW m$^{-2}$ input rates for the Southern Ocean estimated by Wunch (1998). The contribution from the near-inertial band is computed from instantaneous 2-hourly model outputs, time-filtered in the band $\{0.9, 1.15\}f$ following Alford et al. 2012. Near inertial wind work is 1.4 mW m$^{-2}$ for the entire domain and 2.2 mW m$^{-2}$ in its central part (1000 km < y < 2000 km). These values are in agreement with Southern Ocean estimates from drifters (Elipot and Gilles 2009, ~2 mW m$^{-2}$), ocean general circulation models (Rath et al. 2014, ~ 1 mW m$^{-2}$ ) and slab mixed-layer models (Alford 2003, 1-2 mW m$^{-2}$).

## 2.2 Energy diagnostics

Energy diagnostics and precise evaluations of the energy dissipation in the model are essential elements of our study. They are detailed below. The model kinetic energy (KE) equation can be written as follows :

$$\underbrace{\frac{1}{2}\rho_0\partial_t u_h^2}_{KE}=\underbrace{-\rho_0 u_h.\left(u_h.\nabla_h\right)u_h-\rho_0 u_h.w\,\partial_z u_h}_{ADV}\underbrace{-u_h.\nabla_h p}_{PRES}+\underbrace{\rho_0 u_h.D_h}_{\varepsilon_h}+\underbrace{\rho_0 u_h.\partial_z\left(\kappa_v\partial_h u_h\right)}_{\varepsilon_v}+D_{time} \quad (1),$$

where the subscript h denotes a horizontal vector, $\kappa_v$ is the vertical viscosity, $D_h$ the contribution of lateral diffusion processes, and $D_{time}$ the dissipation of kinetic energy by the time stepping scheme, which can be easily estimated in our



simulations since it only results from the application of the Asselin time filter. The dissipation of kinetic energy by spatial diffusive processes is computed as the spatial integral of the diffusive terms $\varepsilon_v$ and $\varepsilon_h$ in equation (1) :

$$E_v = \iiint \underbrace{\rho_0 u_h . \partial_z \left( \kappa_v \partial_z u_h \right)}_{\varepsilon_v} dxdydz = \iiint \left( \rho_0 \kappa_v \frac{\partial u_h}{\partial z} . \frac{\partial u_h}{\partial z} \right) + \iint \left( u_h . \tau_s - u_h . \tau_b \right) dxdy \quad (2),$$

$$E_h = \iiint \underbrace{\rho_0 u_h . D_h}_{\varepsilon_h} dxdydz \quad (3).$$

As mentioned before, we do not specify explicit horizontal diffusion since it is implicitly treated by the UP3 advection scheme we use (see numerical details in Madec 2014). So the term $D_h$ is evaluated at each time step as the difference between horizontal advection momentum tendency computed with UP3 and the advection tendency given by a non diffusive centered scheme alternative to UP3. Two options are the 2nd order and 4th order schemes implemented in NEMO. The 2nd order scheme is non-diffusive but dispersive. The 4th order scheme in NEMO involves a 4th order interpolation for the evaluation of advective fluxes but their divergence is kept at 2nd order, making the scheme not strictly non-diffusive. Although the estimation of UP3 horizontal diffusion depends on the scheme used as a reference we verify in section 5 that the sensitivity of domain averaged $\varepsilon_h$ to the choice of the 2nd or 4th order scheme is much smaller than that resulting from other parameter changes, e.g., small changes in the characteristics of the atmospheric forcing.

## 3. Ocean dynamics under low-frequency forcing

We first examine the dynamics and mean state of the experiment with horizontal resolution 2-km and without storms in order to review the background oceanic conditions within our zonal jet configuration. A snapshot of surface vorticity field (Fig. 3) illustrates the broad range of scale resolved by the 2-km model and the ubiquitous presence of meso- and submesoscale motions, including eddies and filaments. The slope of the annual mean surface velocity spectrum in the meso- and submesoscale range is between $k^{-2}$ and $k^{-3}$. The spectral slope varies seasonally (Fig. 4b), more noticeably in the submesoscale range ($60km > \lambda$), between $k^{-3}$ during summer and $k^{-2}$ during winter (for the meso- and submesoscale range in Fig. 4b, the thin dark red line is superimposed on the thick dark red line). We interpret the increase of submesoscale energy during winter as a direct consequence of enhanced mixed-layer instabilities in response to a deep mixed layer (Fox-Kemper, 2008, Sasaki et al. 2014).

The energy contained at large scale and mesoscale ($k < 5.10^{-5}$ rad m$^{-1}$) decreases with depth as indicated by the spectra at 1000 and 2500 m (Figure 4a). But note that the energy contained in the wavenumber range $5.10^{-5} < k < 6.10^{-4}$ rad m$^{-1}$ (i.e. the range associated with small mesoscale bordering with the submesoscale) is larger at 2500 m compared to 1000m. This is due to an injection of energy at these scales by the rough topography. As shown by instantaneous velocity sections in Fig. 5a,b, the horizontal scales of u and v below 2500 m are much shorter than the typical scale of the upper ocean mesoscale field.



They correspond to the scale of the bathymetry, and are responsible for increased horizontal shear in the deep ocean (Fig. 5e), thereby contributing to the dissipation of the energy imparted by the winds to the mean flow.

Vertical velocity r.m.s. is below 10 m/day over most of the water column except near the bottom (i.e., below 2500 m) where it increases substantially to ~ 100 m/day (Fig. 5c and Fig. 6b). Although flat bottom numerical solutions can also exhibit similar increases (Danioux et al, 2008) the spatio-temporal scales of w near the bottom (e.g, see Fig. 5c) suggest the importance of flow-topography interactions.

The average zonal transport in the reference experiment is ~300Sv. Although the rough bathymetry strongly reduces the transport compared to simulations with flat bottom (that reach ~1000Sv, not shown), the absence of topographic ridge and narrow passages does not allow to obtain the typical observed Antarctic Circumpolar Current transport of ~130-150 Sv (e.g. Cunningham et a. 2003). As discussed in Abernathey et al. (2011), much of this elevated transport can be seen as a translation of the system westward that is not expected to affect our investigation of fine-scale dynamics and its effect on the transverse overturning circulation. The average kinetic energy (KE) exceed 0.05 $m^2$ $s^{-2}$ at the surface (Fig. 6a). Such level of energy is typical of ocean storm tracks of the Southern Ocean (e.g. Morrow et al. 2010).

The clockwise cell of the Eulerian overturning streamfunction $\psi$ (Fig. 7a)[1] illustrates the large scale response to the northward Ekman transport (that acts to overturn the isopycnal) and the irregular return flow in the deep layers due to bottom topography. This transport is largely compensated by an eddy-induced opposing transport, leading to a residual circulation (see e.g. Marshall and Radko 2003). This residual MOC can be computed as the streamfunction $\psi_{iso}$ from the time- and zonal-mean transport in isopycnal coordinates (e.g. Abernathey et al. 2011). In the lightest density classes and northern part of the domain, the counterclockwise cell (negative, driven by surface heat loss) is the signature of a poleward surface flow and equatorward return interior flow, that can be interpreted in terms of mode and intermediate water formation (see the bulge formed by the isothermal layer between the 10 and 12°C isotherms in Fig. 7e). The large clockwise (positive) cell in the center of the domain consists of an upwelling branch along the 1-4°C isotherms and a return flow along the 8-11°C isotherms also contributes to mode water formation. This clockwise cell exhibits a surface protrusion in the temperature range 8-14°C (Fig. 7c) that resembles the upper ocean MOC cell seen in observations (Mazloff et al. 2013) but absent in the semi-idealized experiments with annual mean surface forcings of Abernathey et al. (2011) and Morisson et al. (2011). In our experiments, the upper cell undergoes major seasonal changes (not shown) again in agreement with observations by Mazzlof et al. (2013): clockwise near-surface transport is intensified in boreal summer and fall, when the net heat flux is maximum and warms the upper ocean, enhancing the transformation of the waters toward lighter density classes. This upper cell is thus the result of the seasonal cycle of the surface forcing. Our experiments do not account for the high latitude anticlockwise cell associated with deep water formation because it is of no concern for our purpose. In the 2-km reference case without storms the transport by the main clockwise cell of the MOC streamfunction results in a realistic overturning rescaled value of 18 Sv (Table 2).

---

1 Throughout the paper, Eulerian and residual meridional transports obtained from our 2000 km long channel are multiplied by 10 in order to make them directly comparable to those for the full Southern Ocean whose circumference is ~ 20000 km.



## 4. Single storm effect

As a first step, it is useful to consider a situation in which a single storm disrupts the quasi-equilibrated flow described in the previous section so that high-frequency forcing effects can be more easily identified. The storm is chosen to travel eastward through the center of the domain. The experiment is thoroughly described in section 2 and the ocean spin-down response is analyzed in Figs. 5,8,9 and 10 for a period of 70 days (the storm starts at day 3 and ends at day 7).

### 4.1 NIW generation and propagation

After the passage of the storm, the horizontal currents between the surface and 1500 meters exhibit a layered structure with typical vertical scales of ~ 100-200m (Fig. 5f,g) that contrasts with the homogeneity of the mesoscale currents before the passage of the storm (Fig. 5a,b). The layering is similar to that observed in a section across a Gulf Stream warm core ring by Joyce et al. (2013). It is associated with an increase of the horizontal and vertical shear in the ocean interior (Fig. 5i,j). In agreement with Danioux et al. (2011), we encounter that the storm intensifies the vertical velocities in the whole water column (Fig. 5h). In response to the storm, KE in the upper 100 meters is strongly increased during 5 days (Fig. 8a). An intensification of KE is also observed in the following days at depth below 500 meters, indicative of downward propagation of the energy. A large part of the additional energy injected by the storm occurs in the near inertial range (Fig. 8b) : the space time distribution of the near inertial energy (colors) matches rather well the difference of KE between the experiment with storm and a control experiment without storm starting from exactly the same initial conditions (contours).

   The near-inertial energy propagates downward and its signature can still be observed 60 days after the storm passage with two weak maxima: one at the surface and another centered near 1500 meters. Over the earlier part of the simulation, we find downward energy propagation speeds ~ 25 m day$^{-1}$ in the upper 100 and ~ 90 m day$^{-1}$ between 100 and 1500 meters. These values are higher than the 13 m day$^{-1}$ average propagation speed estimated by Alford et al. (2012) from observations at station PAPA but are within the 10-100 m day$^{-1}$ range estimated by Cuypers et al. (2012) for NIW packets forced by tropical storms in the Indian Ocean. Vertical velocities are generally intensified in the depth range where stratification is weakest but the maximum of r.m.s. vertical velocities qualitatively follows a similar behavior as near inertial KE : it peaks at 2000 m depth a few days after the storm initiation, and then propagates downward the following weeks (Fig. 8c).

   Rotatory polarization of the near inertial waves is useful to separate the upward and downward-propagating constituents of the waves. Rotatory spectra (details of the methodology are given in Appendix B) of the stretched profiles of velocity allow for a separation of the clockwise (CW) and counter-clockwise (CCW) contributions to the energy as a function of time and vertical wavenumber (Fig. 9a,b). Most of the energy is contained in the CW part of the spectra, i.e. most of the energy propagates downward. While the energy directed downward and contained in wavelengths between 1000 and 2000 meters remains strong for about 30 days, the energy at short wavelengths (<500 meters) is rapidly dissipated both for downward and





upward propagating NIWs. The near-inertial KE computed from WKB stretched CW and CCW velocities are shown in Fig. 9c,d. Between days 20 and 30, the KE of CCW waves exhibits a maximum centered around 1500-2000 meters. Because the highest topographic features only reach up to 3000 m depth and also because near-inertial velocities have been WKB scaled we interpret this local maximum as the signature of interior reflection. During the 5 days following the passage of the storm, we notice a slight increase of both CW and CCW KE below 2500 m depth, suggesting NIW generation at the bottom in response to storm forcing. Associated energy levels are limited ($< 10^{-2}$ $m^2 s^{-2}$) and no sign of vertical propagation is observed so this process must be of minor importance, compared to other flow-topographic interactions acting in the same depth range such as lee-wave generation by the balanced circulation (Nikurashin and Ferrari 2010).

Horizontal velocity frequency spectra computed at each depth and averaged over the entire 70-day period of the experiment are shown in Fig. 8f. They exhibit energy peaks at f, 2f and to a lesser extent 3f. The near-inertial and super-inertial peaks are surface intensified but have a signature throughout the water column. Waves with super-inertial frequency arise after a few inertial oscillations and are exited by non-linear wave-wave interactions (Danioux et al, 2008).

### 4.2 Dissipation of the NI energy

We now turn to the identification of the processes (either physical or numerical) that dissipate the kinetic energy imparted by the storm. To this end, the complete energetic balance of the single-storm experiment is compared with that of a control experiment without storm (Fig. 10). After 65 days, the experiment with storm returns to a horizontal kinetic energy level identical to that of the control experiment (Fig. 10a). E-folding time scale for the dissipation of vertically integrated (resp. surface) KE imparted by the storm are ~ 20 days (resp. 5 days). The surface value is consistent with estimates from drifter observations at similar latitudes (Park et al. 2009). The different contributions of the r.h.s. of the kinetic energy equation (Equation 1) that balance the input of energy by the wind work are shown in Fig. 10b. First we note that the cumulated wind work steadily increases after the storm passage (centered at day 5). This is due to a slight strengthening of the large scale eastward surface current in response to the storm (not shown). At day 70, 61.4% of the kinetic energy has been dissipated by diffusive processes in the upper 200m, while 11.1% has been dissipated between 200 and 2000m and 4.3% between 2000m and the bottom (see Table 3). Bottom friction (5.9%) and pressure gradients (5.5%) are also limited sinks for the energy imparted by the storm. The cumulated contributions of horizontal advection and Coriolis forces are small compared to the other terms (<1%). Most of the dissipation due to viscous processes is achieved by vertical processes in the upper 200 meters (80%, Fig. 10c). The maximum contribution of horizontal dissipation is between 200 and 2000 meters where it is stronger than vertical dissipation (Fig. 10c).

Further insights on the distribution of viscous dissipation are obtained by examining the temporal evolution of $\varepsilon_v$ and $\varepsilon_h$ at all depths (Fig. 8d and 8e). It shows that the largest kinetic energy dissipation rates are achieved by $\varepsilon_v$ in the upper 100 m during the 10 days following the storm (Fig. 8d). Interestingly we note the presence of a maximum of $\varepsilon_v$ between 300 and 500 m depth between days 10 and 40, with value of order $10^{-9}$ W $kg^{-1}$. This is due to large shear/dissipation values at depth in and below the core of anticyclonic structures as illustrated in Fig. 5 and confirmed in section 5. At these intermediate depths,



$\varepsilon_h$ and $\varepsilon_h$ are of comparable magnitude. No significant near-bottom increase of $\varepsilon_v$ or $\varepsilon_h$ is found during or after the storm passage in Fig. 8d,e, although NIWs are generated at the bottom in response to the passage of the storm (as seen in the previous section, Fig 9d). The levels of near inertial energy below 2500 m depth remain 2 to 3 order of magnitude lower than those found in the mixed-layer and are not sufficient to significantly increase bottom dissipation.

The time filter contributes to dissipate 14% of the energy imparted by the storm, with dissipation well distributed in the entire water column (Fig. 10d). This dissipation is highly dependent of the time-step used in the simulation (150 seconds) and "Asselin time filter" coefficient (0.1, the default value used in most of the studies with NEMO). In a similar experiment with a time step of 30 seconds the contribution of the Asselin time filter falls to 3.4% (see Table 3) and with an Asselin coefficient of 0.01 it falls to 1.5%. This is coherent with temporal diffusion of the Asselin time filter being proportional to the

product of the Asselin coefficient by the model time step (Soufflet et al., in revision for Ocean Modelling). The temporal diffusion is divided by 5 when using a time step of 30 seconds instead of 150 seconds, and the temporal diffusion is divided by 10 when using a coefficient equal to 0.01 instead of 0.1. In these two sensitivity experiments, the energy that is not dissipated by the temporal filter is dissipated by lateral and vertical diffusion in the entire water column, leading to a vertical distribution of total dissipation (Asselin+$\varepsilon_h$+$\varepsilon_v$) which is similar between experiments (see Table 3).

In terms of meridional distribution, most of the energy is dissipated below the storm track (Fig. 10e). This questions the common hypothesis that a significant part of the energy could be radiated away from the generation area toward lower latitudes (e.g. Garrett 2001, Zhai et al. 2005, Zhai et al. 2009, Blaker et al. 2012, Komori et al. 2008). In our configuration it appears that vertical propagation and dissipation act much faster than horizontal propagation.

**5. Storm effects in quasi-equilibrium**

As a reminder, where and through which mechanisms KE is dissipated, and in particular the extra input of KE associated with storms, is the main focus of our study. The dissipation of the energy imparted by the storms is now investigated in the context of perpetual seasonally varying storm activity where time-averaging can be used to reach statistical robustness. One storm is formed every ten days, travels at constant speed along a given latitude (that changes for each new storm) and has a life-cycle lasting 4 days and composed of 3 phases (mature and linearly growing or decaying). We successively focus on

different related aspects of the simulations energetics : the eddy-kinetic energy (EKE) distribution, the total kinetic energy (KE) balance, vertical distribution of KE dissipation, and the sensitivity of this dissipation to numerics.

**5.1 EKE in the 2-km reference experiments**

The additional input of energy by the storms modifies the levels of kinetic energy in the flow. In the 2-km case without storms, the domain averaged 10-year mean KE computed from zonally averaged velocities is $1.14 \ 10^{-3} \ \mathrm{m^2 \ s^{-2}}$ and the EKE

$(=1/2(u'^2+v'^2)$ where primed velocity anomalies are defined with respect to zonally averaged velocities) is $5.21 \ 10^{-3} \ \mathrm{m^2 \ s^{-2}}$. When storms are included, both quantities increase (mean KE increases to $1.21 \ 10^{-3} \ \mathrm{m^2 \ s^{-2}}$ and mean EKE increases to 5.34

$10^{-3}$ m$^2$ s$^{-2}$). Besides this overall EKE increase, EKE is decreased in the upper 300 m (Fig. 6a). Our interpretation is that this arises owing to the storm reduction of the stratification (Fig. 6d). In turn, this impacts the structure of the vertical modes and the inverse energy cascade in a way that favors a less surface intensified distribution of EKE with storms (Smith and Vallis, 2002). The small enhancement of EKE in the range 1000-2000m in the storm simulation is consistent with this interpretation. There are other impacts of the storms : the r.m.s. of the vertical velocity is increased by one order of magnitude in the whole water column and reaches values of order $10^{-3}$ m s$^{-1}$ (Fig. 6b); the upper 100 meters of the ocean get warmer and less stratified (Fig. 6c,d); and the mixed-layer deepens by $\sim$ 30 m (horizontal lines in Fig. 6c). Obviously, the heat budget is also affected with a +5 W m$^{-2}$ increase of the downward turbulent heat fluxes (Fig. 6e), and air-sea heat fluxes (vertical lines in Fig. 6e).

The ability of near-inertial oscillations to propagate into the ocean interior is affected by the mesoscale field (through the chimney effect, as it will shown in Section 5.3) but is also intimately tied to the shrinking of their horizontal scales so we expect to see non trivial modifications of the KE wavenumber spectra in the presence of storms. Near the surface the storms impact is mainly perceptible at the lowest wavenumbers, the storms forcing scale (Fig. 4a and e), or during summer at submesoscale (Fig. 4b). This larger influence of the storms during summer compared to winter in the submesoscale range is explained by a larger impact of the storms on the mixed-layer depth in summer compared to winter (not shown). During summer, the mixed-layer is shallow (Fig. 2b,g) and sensitive to direct mixing by the storms while during winter the mixed-layer is deeper and its depth is controlled at first order by convective processes with storm passages having a weaker influence. Modifications of the spectral slope ($\sim$ 2.5) by the storms are almost insignificant in the meso-/submesoscale range where surface dynamics energizes the flow, particularly at scales $\sim$ 10 km (wavelength $\sim$ 60 km) and below (Fig. 4a and d). The effect of storms at such fine scales becomes pronounced below $\sim$ 300 m (Fig. 4c), where the surface mode becomes attenuated[2].

At 1000 m where the fine-scale energy associated with the NIW is largest (Fig. 4c) the energy spectrum presents a bulge in the wavenumber range $10^{-4}$ $<$ k $<$ $10^{-3}$ rad m$^{-1}$ that attests of the energy input at such scales. This energy input is larger during winter than during summer (Fig. 4b) in agreement with the storm forcing which is more energetic during winter. Fine-scales energization by the NIW can be seen down to $\sim$ 2500 m (Fig. 4c) where it is confined to lower wavelength than at 1000 m (k $>$ 3. $10^{-4}$ rad m$^{-1}$). Limited signs of a large-scale energy enhancement by the storms can be found at 1000m and 2500m.

## 5.2 KE budget and dissipation in the 2-km reference experiments

Let us first examine in detail the KE balance (Table 4) in the two 2-km reference experiments with and without storms. The KE balance in both experiments are very similar, with overall wind work mainly balanced by the work done by bottom

2 The typical vertical scale H(k) of the surface mode at a wavenumber k is H(k) $\sim$ f / (N k). Using N= $\sqrt{\left(0.2\,10^{-4}\right)}$ (see Fig. 6d) we find H(k=$10^{-4}$)=225 m.



friction (38.9% without storms and 30.5% with storms), pressure work maintaining the system available potential energy (32.2%, 26.0%) and vertical diffusion (23.4%, 33.1%). At the difference of this general balance, and in agreement with results for the single-storm experiment described in section 4, the KE balance also indicates that the additional input of energy provided by the storms (+3.64 mW m$^{-2}$) is balanced at 90% by dissipation (-2.86 mW m$^{-2}$ for horizontal and vertical

dissipation to which one should add the Asselin filter contribution) with pressure work and bottom friction being secondary (respectively -0.18 mW m$^{-2}$ representing a 5% contribution and -0.07 mW m$^{-2}$ representing a 2% contribution). This is in stark contrast with the equilibration of the low-frequency wind work feeding the balanced circulation.

    Now let us focus on the spatial and seasonal distribution of the horizontal and vertical KE dissipation terms $\varepsilon_h$ and $\varepsilon_v$. The vertical distribution of these terms are computed using instantaneous outputs available every 5 days during the last 2-year of

the 2-km runs. This choice of a limited 2-year period is justified given the smallness of the standard deviation of annual mean $\varepsilon$ computed using 20 years of simulation of the experiment with storms (Fig. 11c), e.g., compared to $\varepsilon$ differences we present for different experiments. As stated in section 2, we estimate UP3 intrinsic horizontal diffusivity as the difference between UP3 momentum tendency and the tendency given by a 4$^{th}$ order advective scheme. The alternative use of a 2$^{nd}$ order advection scheme produces very similar estimates of $\varepsilon_h$ (Fig. 11c).

Overall energy dissipation ($\varepsilon=\varepsilon_h+\varepsilon_v$) in the reference experiments is increased by one order of magnitude or more over most of the water column in the presence of storms (Fig. 11a). Exception is found in the lowest 1000 m, where dissipation is always strong because of the interaction of the mesoscale and large scale field with the topography. Without storms, dissipation reaches a minimum of 3. 10$^{-12}$ W kg$^{-1}$ between 1000 and 1500 m depth while the presence of storms increases the level of dissipation to > 10$^{-10}$ W kg$^{-1}$ in this depth range, in agreement with the results for the single-storm experiment (Fig

8).

    The distribution of the dissipation between horizontal and vertical diffusive processes and their respective sensitivity to the energy input by the storms reveals some interesting behavior. First, vertical dissipation dominates in the upper 200 m and (less clearly) below 3000 m, but in between, horizontal processes account for most of the dissipation (Fig. 11b). This is particularly true for the experiment with storms in which $\varepsilon_v$ is systematically less than ¼ of $\varepsilon_h$ below 200m. Second, there is

an increase of horizontal dissipation in the interior in response to the storms (Fig. 11b). This is consistent with enhanced energy at short wavelengths ($\lambda$ < 60km, Fig. 4a,c).

    Since the air-sea heat fluxes and the strength of the storms follow a seasonal cycle, we expect some seasonality of both near surface and interior dissipation. This is examined by comparing $\varepsilon$ profile in summer and winter (Fig. 11d). Values of $\varepsilon$ in the upper 300 meters display large differences between summer and winter, in both experiments with or without storms.

Increased upper ocean energy dissipation during winter is explained by mixed-layer convection in response to surface heat loss. Below 300m, the experiment with storms is the only one that displays seasonal variations of $\varepsilon$, with greatest values during winter. This is coherent with observations by Wu et al. (2011) who observed a seasonal cycle of diapycnal diffusivity



(hence of ε) in the Southern Ocean at depths down to 1800 m, although it reaches somewhat deeper (~ 2500m) in our solutions.

## 5.3 How do mesoscale eddies shape KE dissipation ?

Mesoscale activity is known to affect NIW penetration into the ocean interior (Danioux et al. 2011). In order to clarify the role of mesoscale structures on energy dissipation distribution, an eddy detection method is used to produce composite averages of dissipation, relative to eddy centers. The identification of the eddies is based on a wavelet decomposition of the surface vorticity field (e.g. Doglioli et al. 2007). Following Kurian et al. (2011) a shape test with an error criterion of 60% is used to discard structures with shapes too different from circular. Since the Rossby radius of deformation  varies meridionally within the model domain, composites are built with eddies located between Ly/3 and 2Ly/3, and with diameter larger than 20-km. The barycenter is taken as the center of the eddies and used as reference point to build the composites.

The general distribution of $\varepsilon_h$ and $\varepsilon_v$ within composite eddies (Fig. 12) is in agreement with the vertical distribution of domain averaged ε discussed in the previous section, with increased values of $\varepsilon_h$ and $\varepsilon_v$ near the surface and the bottom. But the composites also highlight the impact of eddies on the distribution of $\varepsilon_h$ and $\varepsilon_v$. As discussed below the distribution of the kinetic energy dissipation within eddies is very different depending on the presence or absence of storms.

Without storms, the distribution of either $\varepsilon_h$ and $\varepsilon_v$ in the upper 1500 meters shows that the border of the cyclones and anticyclones are hot spots of dissipation, while the dissipation at the center of the eddies is weaker than outside (Fig. 12a-d). This was expected since horizontal strain and vertical shear are largest at the edges of eddies and weak within the eddies. Near the bottom, dissipation is increased below the cyclones centers (Fig. 12a,b) and decreased below the anticyclones (Fig. 12c,d), owing to increased near-bottom velocities in cyclones compared to anticyclones (not shown).

In presence of storms (Fig. 12e-h), $\varepsilon_v$ and $\varepsilon_h$ peak at the base of the anticyclones with values higher than $10^{-9}$ W kg$^{-1}$, in qualitative agreement with various observations of NIW trapping at the base of the anticyclones (Joyce et al., 2013; Kunze et al 1995). The largest dissipation is bounded by the contour σ =0.95 f with σ = f + ξ the effective frequency. The compositing highlights the disproportionate importance of anticyclones for NIW dissipation. The total area occupied by the anticyclones that have been picked up by the eddy detection method represents only 2.6% each of the domain area, but concentrate the interior KE dissipation at depth. Between 300 and 1500 meters, 5% of $\varepsilon_h$ and 17% of $\varepsilon_v$ is achieved within identified anticyclones. Conversely, cyclone which statistically occupy a similar area of the model domain are associated with only 4% of $\varepsilon_h$ and 1.9% of $\varepsilon_v$. The statistical importance of anticyclones is further discussed in the conclusion.

## 5.4 Sensitivity tests

How dissipation changes when key physical and numerical parameters are varied is examined below.



*Horizontal resolution.* Energy dissipation is compared in experiments at 20, 5, 2 and 1 km horizontal resolution (Fig. 13). The sensitivity to resolution strongly depends on the considered depth range. Near the surface (0-100m) the dissipation is almost not sensitive to the resolution (Fig. 13a,b and Fig. 14b). This is coherent with the relatively weak variations of the wind work from one resolution to another (Fig. 14a). But below (100-400m), experiments with or without storms show a decrease of $\varepsilon$ when increasing resolution (Fig. 13a,b and Fig. 14c). This decrease is not related to modifications of the wind work (Fig. 14a) and occurs in a depth range affected by upper ocean convection. So it may mostly result from the weakening of the dissipation due to upper ocean convection when resolution increases, as highlighted by the shallowing of the mixed-layer depth [with storms and (*without storms)*: 101m *(93m)* at $\Delta x$=20km, 87m *(67m)* at $\Delta x$=5km, 80m *(59m)* at $\Delta x$=2km, and 68m *(53m)* at $\Delta x$=1km]. This would be in agreement with the re-stratifying effect of the mesoscale and sub-mesoscale flow which become more efficient when resolution increases (e.g. Fox-Kemper, 2008; Marchesiello et al. 2011).

In the depth range 400-3000m, the sensitivity to resolution is highly dependent on the presence or absence of storms. Without storms, a major reduction of dissipation with increasing resolution is noticeable (Fig. 13b). This reduction is of a factor 10 or more in the depth range 400-2000 m, when going from 20 to 1 km resolution (Figs. 13b and 14c,e,f). Concomitantly, the fraction of dissipation due to vertical shear increases because that corresponding to lateral shear drops most rapidly (Fig. 13h). At 1km resolution, it is systematically above 20% down to ~ 2000 m and reaches 50% at 1500 m depth. This contrasts with the run at 20 km where $\varepsilon_v$ is never more than 7% of the total dissipation over the same depth range.

The behavior of interior dissipation with storms is strikingly different. Dissipation changes with resolution are much more modest (in log scale). As mentioned before, dissipation in the upper 100-400 m decreases when going from $\Delta x$=20 km to 1 km (Fig. 14c). Between ~ 400 m and 2000 m, increasing resolution tends to increase dissipation (Figs. 13a and 14d,e). At 20 km the mesoscale field is not well resolved and weaker; therefore the mesoscale near- inertial vertical pump is less efficient in transferring the near inertial energy into the interior. 5 km resolution changes total dissipation significantly (*e.g.*, from 4 $10^{-11}$ W kg$^{-1}$ to 1.2 $10^{-10}$ W kg$^{-1}$ in the depth range 1000-2000m, Fig. 13a and 14f). Changes are modest beyond $\Delta x$=5 km. This is because horizontal dissipation remains nearly unchanged and dominates total dissipation. On the other hand, vertical dissipation exhibits interesting changes in this resolution range. In particular it keeps increasing and so does its overall fraction in total dissipation. Also it develops a weak relative maximum around 300-500 m at 1 and 2km. We relate this maximum to the one seen in dissipation composites for anticyclones (Fig. 12).

Near the bottom important changes also take place when increasing resolution: vertical (resp. horizontal) dissipation decreases (resp. increases) which leads to a slight decrease in dissipation by interior viscous processes. Instead, dissipation by bottom friction increases significantly with resolution (Fig. 14d). We are not sure how to interpret these bottom sensitivities, especially since we do not properly resolve the processes implicated in flow-topography interactions (Nikurashin and Legg, 2011).





*Vertical resolution.*  An experiment with 320 vertical levels has been carried out in which vertical shears (and high-order vertical modes) are better represented than with the reference 50 levels. The vertical thickness of the cells increases from 2 meters at the surface, 5 meters at 500 m depth, 70 meters at 1000 m depth and 180 meters near the bottom. The size of the cells below 2500 meters are equal to the reference experiment so that the local characteristics of flow-topography interactions are unchanged. The overall dissipation ε is increased in presence of storms in the interior in the configuration with 320 vertical levels  (Fig. 13a,b and Fig. 14c-e), indicating that the downward propagation of the NIE is better resolved in the high vertical resolution experiment with more NIE available at depth. Similar increase of ε in the upper 100m in the experiments with and without storms (Fig. 14b) suggests that mixed-layer dynamics are profoundly altered when changing the vertical resolution.

*Advection schemes.* The reference experiment relies on an UP3 advection scheme (Webb et al. 1998). It is compared with three experiments run with three widely used advection scheme : the QUICK scheme which is the default scheme of the ROMS model (Shchepetkin and McWilliams, 2005) and also includes implicit diffusion; a 2$^{nd}$ order centered scheme with a horizontal biharmonic viscosity of $-1.10^9$ m$^4$ s$^{-2}$, and a 2$^{nd}$ order centered scheme with the vector invariant form of the momentum equations (Madec, 2014) with the same horizontal biharmonic viscosity. The implicit dissipation of UP3 and QUICK take the form of a biharmonic operator with an eddy coefficient proportional to the velocity ($A_h = -|u| \Delta x^3 / 12$  with UP3 and $A_h = -|u| \Delta x^3 / 16$ with QUICK). Although QUICK is by construction less dissipative compared to UP3,  ε in both experiments are very similar (Fig. 15a). With or without storms, the 2$^{nd}$ order scheme in flux form (CEN2) or vector invariant form (VFORM) lead to increased ε in the ocean interior with the increase being largest at the bottom (the energy dissipation profiles for the 2$^{nd}$ order and the vector form scheme are so closed that they are superimposed in Fig. 15a). Such distribution of the dissipation changes is obviously related to the choice of a biharmonic coefficient of $-1.10^9$ m$^4$ s$^{-2}$:   characteristic velocities of 1.5 m/s and 2 m/s are required for UP3 and QUICK schemes to match a biharmonic diffusion coefficient of $-1.10^9$ m$^4$ s$^{-2}$. So near the surface where currents are strong the explicit diffusion in the simulations with 2$^{nd}$ order schemes is of same order than the implicit diffusion in QUICK/UP3 simulations, while at depth an explicit biharmonic operator with coefficient $-1.10^9$ m$^4$ s$^{-2}$ overestimates the diffusion compared to UP3/QUICK implicit diffusion. We also note a dissipation increase in the depth range 1000-2000 m when using these schemes in the presence of storms. Sensitivity closer to the surface is much more limited.

*Maximum wind speed.* Stronger winds increase the energy dissipation in the interior (Fig. 15c). Changes in dissipation levels take place from the near-surface down to 2500-3000m which again highlights that near-inertial energy is able to propagate down to such depths. Dissipation changes induced by modifications of the flow-topography interactions would also yield changes in dissipation near the bottom which is not the case, particularly when comparing the 1 N m$^{-2}$ and 1.5 N m$^{-2}$ experiments.


*Storm speed.* The storm speed of the reference experiment was taken as Cs=15 m s$^{-1}$, a value close to the 12 m s$^{-1}$ inferred by Berbery and Vera (1996) in some parts of the Southern Ocean. But this speed is expected to vary from storm to storm and impact the amount of energy deposited into the near-inertial range as several studies have shown in particular in the context of hurricanes (Price 1981, Greatbatch 1983, 1984). The response of the ocean to storms traveling at 20, 15, 10, 5 and 0 m s$^{-1}$ is compared in Fig. 15b with other storm characteristics (including trajectory) remaining unchanged. The storms travel exactly at the same latitude and for the same duration as in the reference experiment with Cs=15 m s$^{-1}$. Above 3000 m depth, energy dissipation increases with storm displacement speed until reaching the threshold of 15m s$^{-1}$ beyond which it reduces slightly. These results are consistent with those of Greatbatch (1984) and in particular NIE is maximized for a storm time scale L/Cs ~ (2 * 500 km) / 15 m s$^{-1}$ ~ 18 hours close to the inertial time scale ($2\pi/f$), with L the scale of the storm. Bottom dissipation is slightly enhanced (from 2 10$^{-9}$ W kg$^{-1}$ to 3 10$^{-9}$ W kg$^{-1}$) when storm speed decreases, presumably as a result of more energy being injected in the balanced circulation when storms move slowly.

More importantly we note that major relative changes in energy dissipation levels occur in the ocean interior as U varies, with one order of magnitude difference or more for storms traveling at 5 m s$^{-1}$ or 0 m s$^{-1}$ compared to storm traveling at 15 m s$^{-1}$ in the depth range 400-2000m. Important changes are also found for U=10 m s$^{-1}$ which further confirms the subtlety of the ocean ringing and its consequences. In particular, note that a 30% increase or reduction of the storm displacement speed has more effect than a 30% reduction in storm strength. It also suggests another possible modus operandi for low-frequency variability in the atmosphere to impact the functioning of the ocean interior through a modification of the storm characteristics such as displacement speed.

## 6. Impact of the storms on the Southern Ocean MOC

KE dissipation and mixing are related in subtle ways. Given the profound modifications of KE dissipation by high frequency winds presented in the previous sections we now assess the influence of the storms on the water mass transformations by examining the MOC sensitivity (Fig. 7). Storms increase the clockwise cell intensity by 3 Sv that is a 16% increase compared to the experiment without storms. This shows that in our experiment the storms contribute efficiently to the strength of the MOC. It is worth mentioning that there are almost no changes in the mean Ekman drift as suggested by the very similar Eulerian overturning streamfunction in the cases with and without storms (Fig. 7a,b).

Both the MOC and the response of the MOC to the storms are sensitive to model horizontal resolution (Table 2). Without storms, the maximum (and scaled) value of the MOC decreases from 20.4 Sv at 20-km to 18.0 Sv at 2-km. This is well related to the decrease of interior (below 100 meters) kinetic energy dissipation with resolution increase in the experiments without storms (Fig. 13b). But when storms are included, the MOC increases with an amplitude that depends on the resolution (+0.3 Sv at 20-km, +1.5 Sv at 5-km, +3.0 Sv at 2-km), leading to transports that are relatively similar between experiments (20.7 Sv at 20km, 20.9 Sv at 5-km and 21.0 Sv at 2-km). Again this is in agreement with the sensitivity of the



kinetic energy dissipation to model resolution : the presence of storms increases the levels of energy dissipation in the interior to a level which remains broadly constant at the different resolutions (Fig. 13a, Fig 14).

The processes that dominate the changes of water mass transformation in the experiments with and without storms can be identified by means of an analysis following Walin (1982), Badin and Williams (2010) and other. Water mass transformation rate G is defined as :

$$G(\rho) = \frac{1}{\Delta\rho} \int D_{air-sea} \, dA - \frac{\partial D_{diff}}{\partial \rho},$$

with $D_{diff}$ the diffusive density flux and $D_{air-sea}$ the surface density flux given by

$$D_{air-sea} = -\frac{\alpha}{C_p} Q_{net},$$

with $Q_{net}$ the net surface heat flux, $C_p$ the heat capacity of the sea water, $\alpha$ the thermal expansion coefficient of sea water, and $\Delta\rho$ the density integration interval. The diapycnal volume flux is directed from light to dense waters when G is positive. The computation of the different terms is achieved following the technical details provided in Marshall et al. (1999) with density bin $\Delta\rho$ of 0.1 kg m$^{-3}$. For ease comparison with previous results, the diagnostics are performed in temperature space. As for momentum diffusion, the horizontal diffusion of temperature is computed as the difference between UP3 temperature tendency and the tendency given by a 4$^{th}$ order centered scheme.

In the 2-km experiments without storms, the transformation by air-sea fluxes is mainly from dense to light waters and peaks at -13 Sv near 6ºC (Fig. 16b, again the values here are scaled to the full Southern Ocean). At this temperature, the transformation by diffusive processes only reaches a modest -1 Sv (Figure 16c) and the total transformation rate (~ -14 Sv) is consistent with the 14.5 Sv of meridionally averaged MOC transport centered at 6ºC (not shown). The transformation by diffusive fluxes has two extrema near 4ºC and 12ºC, which correspond to temperatures where convection is more active as suggested by the isolines of cumulative distribution of mixed-layer depth in Fig. 7f or by the seasonal cycle of the mixed-layer depth in Fig. 2b.

Overall, storms increase both the transformation by air-sea fluxes (~+3Sv or +25% at 6ºC) and diffusive fluxes (~+2Sv or +130% at 4ºC), leading to a ~+3 Sv total increase of water mass transformation is the isotherm range 4-8ºC (Figure 16a) that is consistent with the +3 Sv strengthening of the main clockwise cell of the MOC. In presence of storms, the contributions from lateral and vertical diffusion are almost equal (Fig. 16d), while without storms lateral diffusion dominates the water mass transformation (Fig. 16c). The fraction of transformation achieved below 300 m depth is very weak indicating that most of the diffusive transformation process takes place in the near surface (Fig. 16c,d). On the other hand, an important caveat is that only ~ 20% of the energy dissipated below 300 m is properly connected to mixing (through the k-epsilon submodel).



## 7. Discussion

### 7.1 Model realism and limitations

The realism of model dissipation is difficult to evaluate against observations of dissipation rates because of spatial variability and temporal intermittency in nature (see for example the longitude dependence of the dissipation rate found by Wu et al. (2011) in the Southern Ocean; variability at finer scale is also important). With storms, *mean* interior dissipation values at the highest resolution are in the range 1-10 $10^{-10}$ W kg$^{-1}$ depending on exact depth above 2000 m and season. Such values are consistent with estimates from microstructure measurements (Waterman et al. 2013, Sheen et al. 2013) or from release and tracking of dye at mid-depth (Ledwell et al. 2011). However they are on the lower end of the ARGO estimates of Wu et al (2011).

A source of uncertainty in comparing our simulations to observations is that we lack internal gravity waves generation by tides and we also misrepresent the interaction between the geostrophic flow and bottom topography. Both of these processes should significantly contribute to near-bottom dissipation enhancement and their consequences around mid-depth may not be negligible. Cabbeling and thermobaricity are other indirect sources of mixing that are not taken into account in our study.

Assuming that Wu et al. (2011) estimates in regions with smooth bathymetry primarily reflect dissipation of wind-input energy we can nonetheless make two important quantitative remarks. The vertical structure of storm energy dissipation in our simulations is qualitatively consistent with their observations: we find a factor 5-6 reduction in dissipation from 400 m to 1800 m depth as they approximately do (their figure 3). Model seasonal variations in ε also agree (note that we infer seasonal changes of ε in Wu et al. 2011 from changes in diapycnal diffusivity, assuming that subsurface stratification does not vary between seasons). Model (resp. observations from Wu et al, 2011) winter to summer ε ratios decrease from ~ 2 (resp., ~ 1.8) in the depth range 300-600 m to 1.6 (resp., ~ 1.4) in the depth range 1300-1600 m. These numbers agree within the error bars associated with Wu et al's observations. On the other hand, it is plausible that the slightly weaker seasonal cycle systematically found in the observations arises from dissipative contributions due to processes other than wind. The respective roles of wind-input and that of a distinct non-seasonally variable process on dissipation could in principle be separated but model uncertainties and limitations should also be kept in mind.

Near the bottom our simulations generate dissipation at levels that are essentially unaffected by synoptic wind activity (although this is less true when storms travel slowly). ε reaches ~5. $10^{-9}$ W kg$^{-1}$, a value which is not overly affected by numerical resolution and turns out to be close to the values measured or inferred near rough topography (Waterman et al. 2013, Sheen et al. 2013). This being said, important reorganizations in the bottom 500 m from vertical to horizontal dissipation as horizontal resolution increases suggest cautiousness. So does the unrealistic representation of internal lee-wave processes.



## 7.2 Energy pathways

Results by Nikurashin et al. (2013) suggest that the bulk of the large scale wind power input in the Southern Ocean is dissipated at the bottom by the interaction of the mesoscale eddy field with rough (small scale) topography. Our simulations also show high energy dissipation at the bottom, but instead of as in the rough experiment described in Nikurashin et al.

(2013), for which most of the energy imparted by the wind is balanced by interior viscous dissipation, the wind input in our 2-km experiment without storms is balanced by bottom friction (38.9% associated with unresolved turbulence in the bottom boundary layer), pressure work (32.2%) and interior viscous dissipation (23.4%). This points out that we are not exactly in the same regime than the one described in Nikurashin et al. (2013). This is probably related to low roughness of our experiments compared to the rough experiment in Nikurashin et al. (2013).

Using a global high-resolution model, Furuichi et al. (2008) estimate that 75-85% of the global wind energy input to surface near inertial motions is dissipated in the upper 150 m. Similarly, Zhai et al. (2009) analysing a global 1/12° model found that nearly 70% of the wind-induced near-inertial energy at the sea surface is lost to turbulent mixing within the top 200 m. Our results are noticeably different: in our high-resolution simulations only ~65-70% of the overall energy imparted by the storm is dissipated in the upper 200m (65% in the one storm experiment, see Table 4; 70% in the multiple storm

experiment, not shown). Note though that, in contrasts to Furuichi et al. (2008) who base their estimate on the near-inertial response of the wind energy input, we do not separate the balanced and unbalanced response to the storms. A substantial part of the additional wind work imparted by the storms is not near-inertial, as revealed by the 1.4 mW m$^{-2}$ near-inertial wind work in the experiment with storms, which is only a fraction of the +3.6 mW m$^{-2}$ total wind work increase compared to the experiment without storms. Since the balanced response to the storms does not follow the same pathway toward dissipation

(see below), such differences between our results and Furuichi et al. (2008) are not unexpected.

Finally, our experiments provide an interesting perspective on the dissipation of the energy associated with the slow versus NIW part of the flow. The ways the energy imparted to the ocean by high and low frequency winds are balanced differ markedly as one may have expected. Wind work imparted by the storms is mainly balanced by viscous dissipation (> 80%), mainly in the upper ocean and to a lesser extent in the interior. Bottom friction (~ 5%) and pressure work (~5 %) play a

25 minor role while these two terms are key in the equilibration of the low-frequency part of the circulation (note that the loss term associated with pressure gradient forces represents the potential energy source due to Ekman pumping). Perhaps more surprisingly, total interior dissipation in the simulation with and without storms present distinct sensitivities with respect to resolution. As horizontal/vertical resolution increases storm energy dissipation tends to diminish within a few hundred meters below the mixed layer base but increases farther down. Conversely, dissipation of the balanced circulation sharply

decreases with increasing resolution over a broad range of depth in the ocean interior, from below the mixed layer down to 3000 m depth. It is also the situation where convergence is least clear in the range of resolutions that we explore. Even Δx=1km resolution may still be insufficient to adequately resolve fine-scale dissipative processes affecting the balanced flow (Vanneste 2013). In any event and far from topographic features, dissipation of the balanced flow which is robustly 1 to 2





order of magnitude smaller than dissipation of the NIE below 300 m depth is unlikely to have a substantial effect on diapycnal mixing in the Southern Ocean interior.

## 8. Conclusions

Kinetic Energy (KE) dissipation and its effect on ocean mixing are a subject of intense research. The aim of this study is to
investigate the fate and the overall impact of the energy imparted by the storms in the Southern Ocean. The set of semi-idealized numerical simulations we use to this end allow us to explore and to identify the limitations faced by the general/regional ocean modelling community in the numerical representation of these processes. We also provide an additional perspective on the MOC sensitivities (to high frequency winds) in a semi-idealized representation of the Southern Ocean that shares important characteristics with the ones used in Abernathey et al. (2011), Morrison and Hogg (2013; MOC
sensitivity to the mean wind stress) or Morrison et al. (2011; sensitivity to surface buoyancy forcing).

The main oceanic response to storm forcing involves the generation and downward propagation of NIWs. While ~60% of the energy imparted by the storms is dissipated in the upper 200 meters, a substantial part propagates and dissipates at deeper depths (~ 20-30%). The NIWs that penetrate downward have short horizontal wavelengths ($\lambda < 60km$), high vertical shear and horizontal strain variance, contributing to their dissipation before they reach the bottom.

In our simplified simulations near-inertial oscillations are the dominant source of mixing down to 2000-2500 m depth. Our model results also confirms the conclusions of several previous numerical and observational studies: atmospheric synoptic variability and its associated internal energy wave activity generation is required to explain the levels of mixing observed in the interior ocean away from rough bathymetric features. This additional input of energy becomes critically important as the resolution increases and viscous dissipation of the balanced circulation vanishes (without storms a two order of magnitude
reduction of interior dissipation is found when going from $\Delta x=20$ km to $\Delta x=1km$). The inclusion of storms lead to comparatively minor sensitivities of interior dissipation to model resolution. This has profound consequences on the MOC sensitivity to model horizontal resolution : while without storms the strength of the clockwise cell of the MOC decreases when resolution increase (also observed in Morrison and Hogg 2013), the introduction of storms tends to level off the differences between resolution and to produce a slight increase of the MOC with increasing resolution (Table 2).

We have shown that anticyclones play a disproportionate role as a conduit to the interior ocean dissipation. This could certainly be anticipated from the several studies describing the presence and dissipative fate of NIW packets in anticyclonic structures. We are able to characterize this statistically. We found that between 300 and 1500 meters, 17% of the dissipation achieved by vertical processes occurs within identified anticyclones (versus 2% within identified cyclones). This estimate is a conservative figure because we use a stringent eddy identification procedure.

Even with the storms included, dissipation below 200-300 m is too modest to substantially influence water mass transformation (section 5). This result should however be considered cautiously. Increased resolution (particularly horizontal) beyond the range we explored may lead to further enhancements of dissipation in the depth range 200-500m.





More importantly perhaps, horizontal dissipation (which results from implicit numerical diffusion in the advection of momentum) is dominant below the mixed layer and its effects on diapycnal mixing may not be adequately represented. Indeed, it does not contribute to the calculation of vertical mixing of temperature and its connection with horizontal mixing (also resulting from implicit numerical diffusion) is unknown[3]. The relation between energy dissipation and mixing is a subject of intense research. Ground-truth exists from Direct Numerical Simulations (DNS) or lab experiments (Shih et al, 2005, Ivey et al, 2008) but their utilization is not straightforward here because of the large scale gap with our simulations in terms of resolved length scales (our ~ 1km horizontal resolution places us several orders of magnitude away from the isotropic regime).

The effect of storms is obviously most significant in the upper ocean. A Walin analysis highlights this role and the consequences on large scale ocean dynamics. In our simulations storms significantly modify the vertical buoyancy flux, air-sea heat fluxes (which are interactive) and MOC intensity (+16%). Although the settings have differences an instructive comparison consists in estimating the change in mean wind stress required to increase the upper MOC cell (the only one we simulate) by 16% in the sensitivity experiments carried out by Abernathey et al (2011). Their figure 5 indicates a change from 0.20 N m$^{-2}$ to ~0.23 N m$^{-2}$ (+15%) is needed when interactive air-sea flux are used. This further confirms the importance of synoptic winds. The effect of storms expressed in terms of change in net air-sea heat fluxes is less dramatic (+5 W m$^{-2}$) and well within uncertainties (Wainer et al, 2003). On the other hand, the fluctuations of heat fluxes due to storms have not been considered in our study and their impact should be further investigated.

Important conclusions of this study also concern the numerical and physical sensitivities of the NIE fate. Our analyses and sensitivity runs highlight the effect of the Asselin filtering, of the numerical scheme employed for advection, of numerical resolution, horizontal and to a lesser extent vertical. Although $\varepsilon$ changes with horizontal resolution tend to level off when approaching $\Delta x$=1 km, a more subtle lack of convergence is patent. Most importantly, the respective contributions of horizontal and vertical dissipations to $\varepsilon$ still exhibit major changes between $\Delta x$=2 km and to $\Delta x$=1 km, mainly in the depth range 200-500 m. The reason why this may be of concern is that vertical and lateral dissipation have a priori very different consequences in the model, in ways that are difficult to reconcile with the isotropy of microscale turbulence measured in the real ocean. In the model, vertical dissipation is an essential component of the vertical turbulent closure and modulates diapycnal mixing. Although lateral dissipation may also be accompanied by diapycnal mixing (near fronts), existing ocean models have not been widely evaluated or tuned in this regard. Ongoing efforts are aimed at reducing lateral diapycnal diffusion in OGCMs but it is unclear down to which level this should be pursued. The tendency found over the range of $\Delta x$ explored in this study suggests a robust $\varepsilon_v$ increase to the detriment of $\varepsilon_h$ at depths between 200 and 500 m. The strength of the diapycnal mixing that takes place in this ocean range is important as demonstrated by the MOC sensitivity analysis in section 6. Further efforts to approach convergence and diminish grid anisotropy for problems resembling to the one studied here would be needed.

---

3 Note that in the case where diffusion and viscosity operators and coefficients are explicitly prescribed no consistency between KE dissipation and horizontal mixing of temperature is enforced either.





The modifications of the Southern Ocean atmospheric circulation have motivated many studies on the response of the ACC and Southern Ocean overturning to increase in mean wind stress. But besides zonal wind strengthening, changes are also observed in the storm track activity (see the review by Ulbrich et al. 2009). The evolution during the last 50 years consists in a concomitant decrease of the overall number of Southern Ocean cyclones and increase of their strength. This tendency is

expected to continue under warming climate. Alford (2003) estimated a 25% increase from the 1950's of global power input to inertial motions. The subtleties of interior mixing forced by high-frequency winds, as highlighted by our study, add to the list of challenges awaiting eddy-permitting/eddy resolving climate models.

**Acknowledgments**

This study has been supported by CNRS and has been founded by the French ANR project SMOC. Supercomputing facilities were provided by PRACE project RA1616 and GENCI project GEN1140. The authors wish to thank Y. Cuypers, E. Pallàs-Sanz, L. Debreu, F. Lemarié and P. Marchesiello for useful discussions. Interactions with the Communauté de Modélisation (COMODO, ANR funding) are also acknowledged. We are grateful to one anonymous reviewer for helpful comments on the manuscript.

**Appendix A : Wind forcing strategy**

The Southern Ocean storms are represented as cyclonic Rankine vortices :

$$\tau_\theta = \tau_{max} \frac{r}{R} \quad \text{if } 0 \leq r < R$$

$$\tau_\theta = \tau_{max} \frac{R}{r} \quad \text{if } R \leq r$$

with $\tau_{max}$ the maximum wind stress, R the radius of the vortex core (300 km). $\tau_\theta$ is set to zero for r > 900 km. $\tau_{max}$ is

modulated by a sinusoidal seasonal cycle so it varies from $\tau_{max0}/2$ during austral summer and $\tau_{max0}$ during austral winter, with $\tau_{max0} = 1.5$ N m$^{-2}$. Each vortex forms and vanishes at the same latitude (no meridional displacement) but the latitude of formation varies following a Gamma distribution similar to the meridional distribution inferred from cyclones tracks in Patoux et al. (2009), with most of the cyclones located between 50ºS and 70ºS. The distribution follows a cycle which repeats each 10 years. Life time of the storms is computed such that one cyclone travel the 2000 km zonal extension of the

domain with full strength (~ 2 days). This strategy leads the storm to wrap around itself during its decaying phase, but note that this only affect a limited portion of the domain. One storm is formed each ten-days. The cyclones form and vanish linearly in one day, and travel eastward at a speed Cs of 15 m s$^{-1}$ in the reference experiment. Cyclone position and associated winds are recomputed at each time step.



The wind-stress in the experiments without storms is purely zonal :

$$\tau_b = \tau_0 \sin\left(\pi y / L_y\right) \ ,$$

with $\tau_0 = 0.15$ N m$^{-2}$. In order to have exactly the same 10-year mean wind stress between experiments with and without storms, the averaged residual wind due to the storm passages is removed to the background wind stress in the experiment with storms.

**Appendix B : Rotatory spectra**

The computation of rotatory spectra follow the methodology described in Leaman and Sanford (1975) and other (e.g., Alford et al, 2012). First the near inertial part of the velocities $u^{niw}$ are obtained by filtering the velocity components in the near-inertial band $\{0.9, 1.15\}f$. These velocities are then normalized at each depth as follows :

$$u_n^{niw}(z) = u^{niw}(z) / \sqrt{N(z)/N_0},$$

where $u_n(z)$ is the normalized velocity, $u(z)$ is the band-pass filtered velocity, $N(z)$ is the Brunt-Väisälä frequency and $N_0$ is the vertical average of $N(z)$. The velocity are then Wentzel–Kramers–Brillouin (WKB) stretched according to $dz' = N(z)/N_0\ dz$ with $z'$ the stretched and $z$ the unstretched coordinates.

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



| Name | Δx | Nb vert. levels | Dt (Asselin coefficient) | Horiz. adv scheme | Storms | Storm speed (m/s) | $T_{max}$ (N m-2) |
|---|---|---|---|---|---|---|---|
| *Sensitivity to horizontal and vertical resolution* | | | | | | | |
| 20-km-nostorm | 20km | 50 | 1200s (0.1) | UP3 | no | | |
| 20-km-storms | 20km | " | 1200s (0.1) | " | yes | 15 | 1.5 |
| 5-km-nostorm | 5km | " | 300s (0.1) | " | no | | |
| 5-km-storms | 5km | " | 300s (0.1) | " | yes | " | " |
| 2-km-nostorm | 2km | " | 150s (0.1) | " | no | | |
| 2-km-storms | 2km | " | 150s (0.1) | " | yes | " | " |
| 1-km-nostorm | 1km | " | 60s (0.1) | " | no | | |
| 1-km-storms | 1km | " | 60s (0.1) | " | yes | " | " |
| 2-km-nostorm_Z320 | 2km | 320 | 50s (0.1) | " | no | | |
| 2-km-storms_Z320 | " | 320 | 50s (0.1) | " | yes | " | " |
| *Sensitivity to horizontal advection scheme* | | | | | | | |
| 2-km-nostorm_QUICK | " | 50 | 150s (0.1) | QUICK | no | | |
| 2-km-storms_QUICK | " | " | 150s (0.1) | QUICK | yes | " | " |
| 2-km-nostorm_CEN2 | " | " | 100s (0.1) | CEN2 | no | | |
| 2-km-storms_CEN2 | " | " | 100s (0.1) | CEN2 | yes | " | " |
| 2-km-nostorm_VFORM | " | " | 100s (0.1) | VFORM | no | | |
| 2-km-storms_VFORM | " | " | 100s (0.1) | VFORM | yes | " | " |
| *Sensitivity to storm characteristics* | | | | | | | |
| 2-km-storms_C0 | " | " | 150s (0.1) | UP3 | yes | 0 | " |
| 2-km-storms_C5 | " | " | " | " | yes | 5 | " |
| 2-km-storms_C10 | " | " | " | " | yes | 10 | " |
| 2-km-storms_C15 | " | " | " | " | yes | 15 | " |
| 2-km-storms_C20 | " | " | " | " | yes | 20 | " |
| 2-km-storms_TAU-1 | " | " | " | " | yes | 15 | 1 |
| 2-km-storms_TAU-1.5 | " | " | " | " | yes | 15 | 1.5 |
| 2-km-storms_TAU-3 | " | " | " | " | yes | 15 | 3 |
| *One storm experiments* | | | | | | | |
| 2-km-onestorm_A | " | " | 150s (0.1) | " | yes | 15 | 1.5 |
| 2-km-onestorm_B | " | " | 30s (0.1) | " | yes | 15 | 1.5 |
| 2-km-onestorm_C | " | " | 150 (0.01) | " | yes | 15 | 1.5 |

**Table 1.** Summary of numerical experiments.





|  | 20-km | 5-km | 2-km |
|---|---|---|---|
| **No storm** | 20.4 Sv | 19.4 Sv | 18.0 Sv |
| **Storms** | 20.7 Sv | 20.9 Sv | 21.0 Sv |

**Table 2.** Maximum of the clockwise cell (as in the context of Figure 7) of the overturning streamfunction $\Psi$iso (Sv) averaged between y=2000km and y=2500km. The streamfunctions have been computed using ten years of 5-day average outputs from equilibrated experiments. Model transports have been multiplied by 10 in order to scale them to the full Southern Ocean.



| | Δt 150s / Asselin 0.1 (horiz.,vert.) | Δt 30s / Asselin 0.1 (horiz.,vert.) | Δt 150s / Asselin 0.01 (horiz.,vert.) |
|---|---|---|---|
| $\mathcal{E}h+\mathcal{E}v$ 0-200m | 61.4% (3.1%, 58.3%) | 63.7% (4.3%, 59.4%) | 64.4% (4.6%, 59.8%) |
| $\mathcal{E}h+\mathcal{E}v$ 200-2000m | 11.1% (9.3%, 1.8%) | 14.9% (12.6%, 2.3%) | 15.6% (13.2, 2.4%) |
| $\mathcal{E}h+\mathcal{E}v$ 2000-4000m | 4.3% (2.9%, 1.4%) | 5.8% (3.9%, 1.9%) | 5.9% (4.1%, 1.8%) |
| **Bottom friction** | 5.9% | 6.6% | 6.7% |
| **Coriolis** | - 0.1% | - 0.2% | 0.2% |
| **Advection** | - 0.7% | - 0.6% | -0.6% |
| **Pressure** | 5.5% | 5.2% | 5.2% |
| **Asselin time filter** | 14% | 3.4% | 1.5% |
| **DKE/Dt** | -1.4% | 1.2% | 1.1% |
| **Residual** | 0.% | 0.% | 0.% |
| *Total dissipation ($\mathcal{E}h+\mathcal{E}v$+Asselin)* | | | |
| **Full water column** | 90.9% | 87.3% | 88.0% |
| **0-200m** | 65.4% | 64.7% | 64.5% |
| **200-2000m** | 19.5% | 16.5% | 17.1% |
| **2000-4000m** | 6% | 6.1% | 6.4% |

**Table 3.** Cumulated energy dissipation at day 70 (see Fig. 10) relative to a reference experiment without storm, for three single-storm experiments with different time step and Asselin time filter coefficient. Results for the reference experiment described in Fig. 5,8-10 are shown in the first column.



|  | 2-km [mW m⁻²] | 2-km + STORMS [mW m⁻²] | Differences [mW m⁻²] |
|---|---|---|---|
| **DKE/Dt** | -0.1 | 0.02 | |
| **Wind work** | 12.43 | 16.08 / | +3.65 |
| **Vertical dissipation** | -2.90 (23.4%) | -5.32 (33.1%) | -2.42 (66.3%) |
| **Horizontal dissipation** | -0.65 (5.2%) | -1.09 (6.8%) | -0.44 (12.0%) |
| **Pressure work** | -4 (32.2%) | -4.18 (26.0%) | -0.18 (5%) |
| **Bottom friction** | -4.84 (38.9%) | -4.91 (30.5) | -0.07 (2%) |
| **Advection** | -0.03 (0.2%) | 0.01 (0.1%) | -0.02 (0.5%) |
| **Coriolis** | -0.08 (0.6%) | -0.06 (0.4%) | -0.02 (0.5%) |
| **Asselin time filter** | -0.01 (0.1%) | -0.45 (2.8%) | -0.44 (12%) |
| **Residual** | 0 | 0 | 0 |

**Table 4.** Two-year mean KE balance (mW m⁻²) averaged over the entire domain for the 2-km reference experiments with (left) and without storm (right). The percentages give the fraction of total wind-work that is balanced by the terms of the KE equation. The second series of numbers and percentages in the storm column refers to the storm - no storm differences.




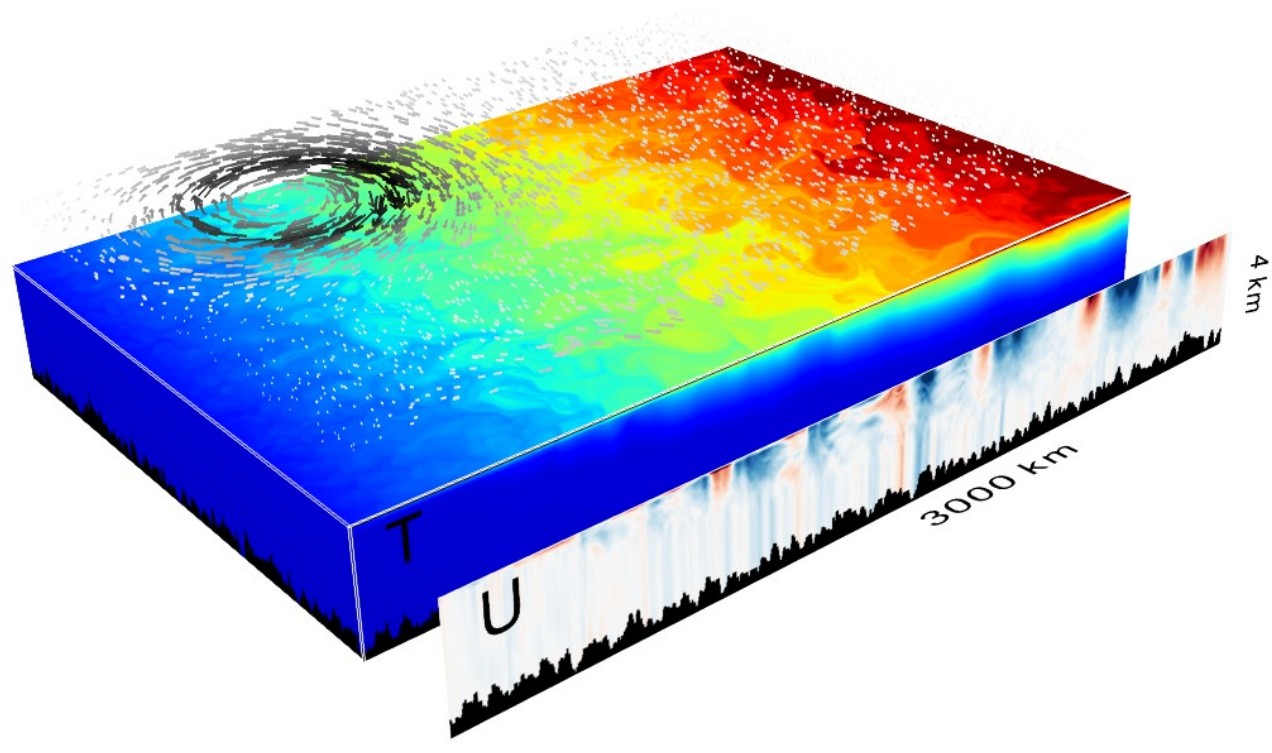

**Fig. 1** 3D representation of instantaneous temperature (rectangular box, color scale ranges from 0 to 20°C) and zonal velocity (vertical section) for the reference simulation at 2-km after 30 years. The domain is a 2000 km long - 3000 km wide reentrant channel. The configuration represents the Southern Ocean between 40ºS and 70ºS. Average ocean depth is 3500m with irregular bottom topography, which limits the ACC transport and tends to enhance deep mixing. At the surface, synoptic storms are included in the forcing. They generate NIWs whose signature is visible in the velocity section, as a layering of the mesoscale structures.



**Fig. 2.** Seasonal cycle of zonally averaged SST (a-f, ºC), mixed-layer depth (b-g, meters), net air-sea heat flux (c-h, W m$^{-2}$), and the solar (d-i, W m$^{-2}$) and non solar (e-j, W m$^{-2}$) components of the air-sea heat flux. Climatological seasonal cycles are built from observations (left column) and model outputs and forcing. Observations include OAFlux products (Yu et al. 2007) for the period 1984-2007 and de Boyer Montégut (2004) mixed-layer depth climatology. Model data are from the last 10 years of the 2-km reference simulation without storms. In both model and observations, the mixed-layer depth is computed with a fixed threshold criterion (0.2°C) relative to the temperature at 10 meters.





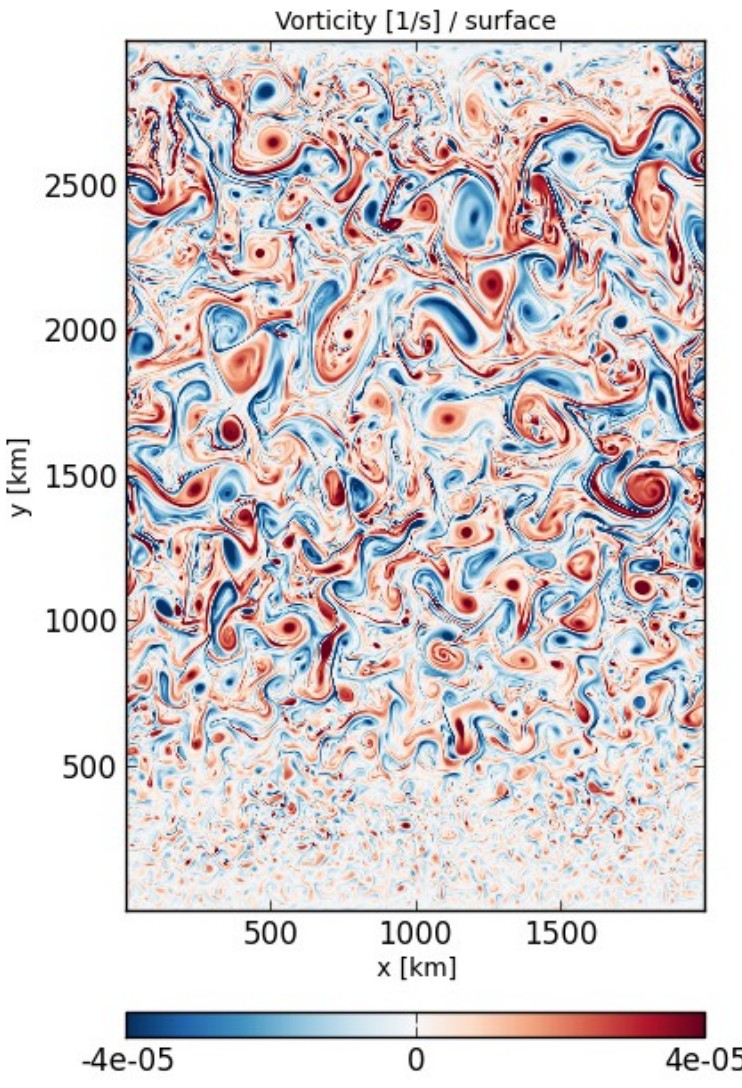

**Fig. 3.** Surface vorticity snapshot (s⁻¹) over the entire model domain at day December 31th of year 39 from the 2-km horizontal resolution experiment without storms.





**Fig. 4.** Horizontal velocity variance in the 2-km reference experiments with and without storms. (a) Kinetic energy power spectra as a function of wavenumber (rad m$^{-1}$) at 0, 1000 and 2500 m depth. (b) Seasonal (summer is defined as December-January-February and winter as June-July-August) kinetic energy power spectra at 0 and 1000 m depth. Spectra are built using instantaneous velocity taken each 5 days of the last 2 years of the 2-km simulations. Kinetic energy contained in the wavelength ranges $\lambda < 60$km (c), $60$km $< \lambda < 600$km (d), and $\lambda > 600$km (e) as a function of depth. In (b) and for wavenumber above $5 \cdot 10^{-5}$ rad m$^{-1}$, the winter surface spectra with and without storms (dark red thin and thick lines) are superimposed, as well as the summer and winter 1000-m spectra without storms (light and dark green thin lines).



**Fig. 5.** Model snapshots of a 2-km simulation at a mesoscale eddy location 2 days before (top) and 17 days after (bottom) the passage of a storm : (a,f) zonal velocity (m s$^{-1}$), (b,g) meridional velocity (m s$^{-1}$) (c,h) vertical velocity (m s$^{-1}$), (d,i) vertical shear (s$^{-2}$) and (e,j) horizontal strain (s$^{-2}$). Snapshots after the passage of the storm (e-h) are taken 50 km eastward in order to account for the advection of the core of an anticyclonic mesoscale eddy. Isotherm are shown in the left panels with contour intervals of 1.25°C from 2.5 to 10°C. Before the passage of the storm the simulation has been equilibrated without high-frequency forcing, so the solution at day -2 is free of wind-forced NIWs. The snapshots shown here correspond to day 2 and 22 in the time axis of Fig. 8.

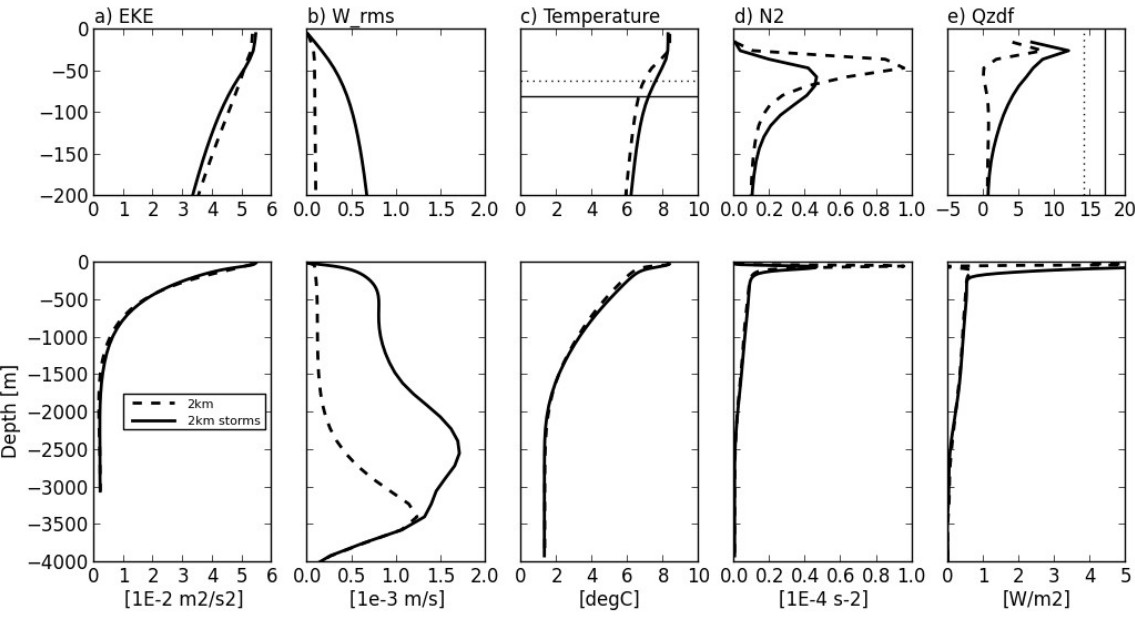

**Fig 6.** Vertical profile of the 2-km experiments with and without storms averaged over a period of 10 years and between Ly/3 a 2Ly/3 with Ly the meridional length of the domain : (a) eddy kinetic energy (m$^2$ s$^{-2}$), (b) r.m.s of the vertical velocity (m s$^{-1}$), (c) temperature (ºC), (d) stratification (s$^{-2}$) and (e) vertical turbulent heat flux (W m$^{-2}$). The eddy kinetic energy is computed from anomalies to the zonal mean. Dashed lines are for the experiment without storms. In (c) the horizontal lines indicate the mean position of the mixed-layer base and in (e) the vertical lines show the average net air-sea heat flux (W m$^{-2}$).




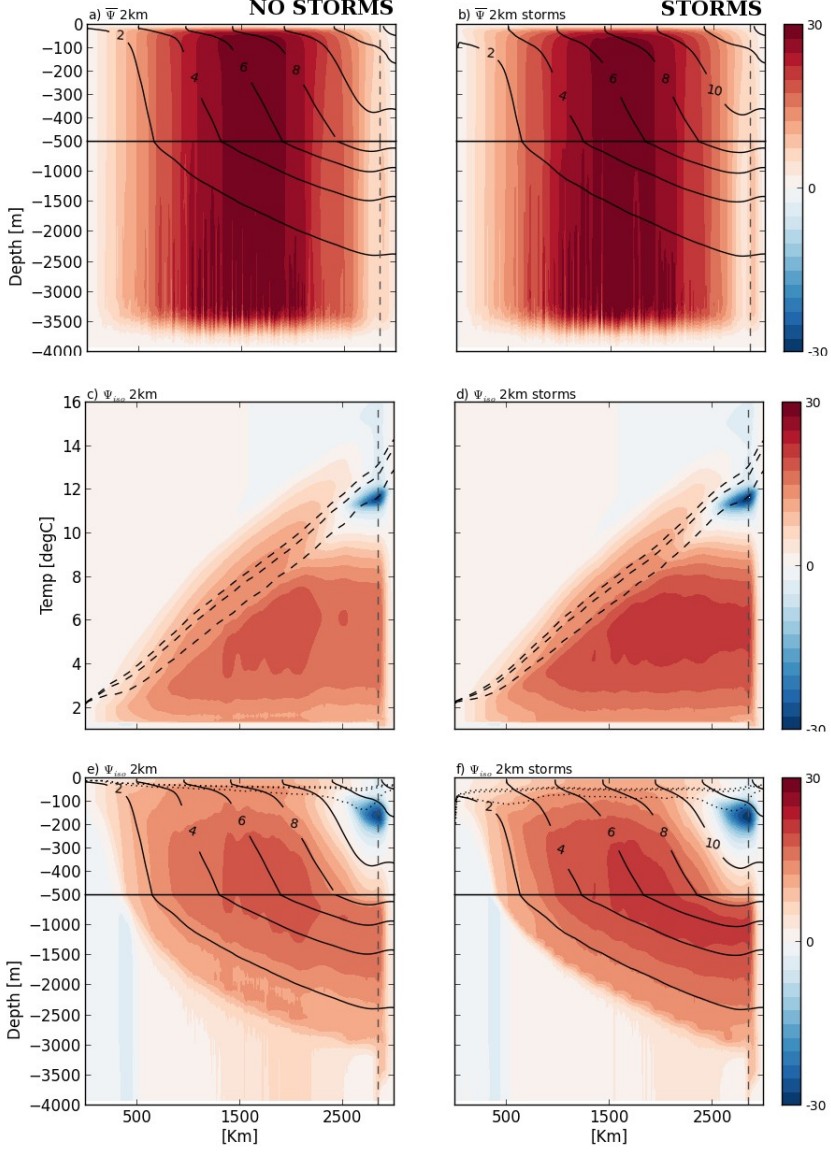

**Fig. 7.** Eulerian mean streamfunction ψ (a,b), MOC streamfunction diagnosed in isopycnal coordinates (c,d) and projected back to depth coordinates (e,f), from 10-years long 2-km equilibrated simulations with (right) and without storms (left). Units are Sv and the contour interval is 0.25 Sv. Temperature contours corresponding to 2, 4, 6, 8, 10, 12 and 14°C are indicated in (c,d). Positive cells are clockwise. The dotted lines represent the 10%, 50%, and 90% isolines of the cumulative surface temperature distribution (following Abernathey et al. (2011) and which tells how likely a particular temperature is to be found at the surface and thus be exposed to diabatic transformation ) (in a,b), and cumulative mixed-layer depth distribution (in c,d). The vertical dashed line at y=2850km represents the limit of the northern boundary damping area. Model transports have been multiplied by 10 in order to scale them to the full Southern Ocean.





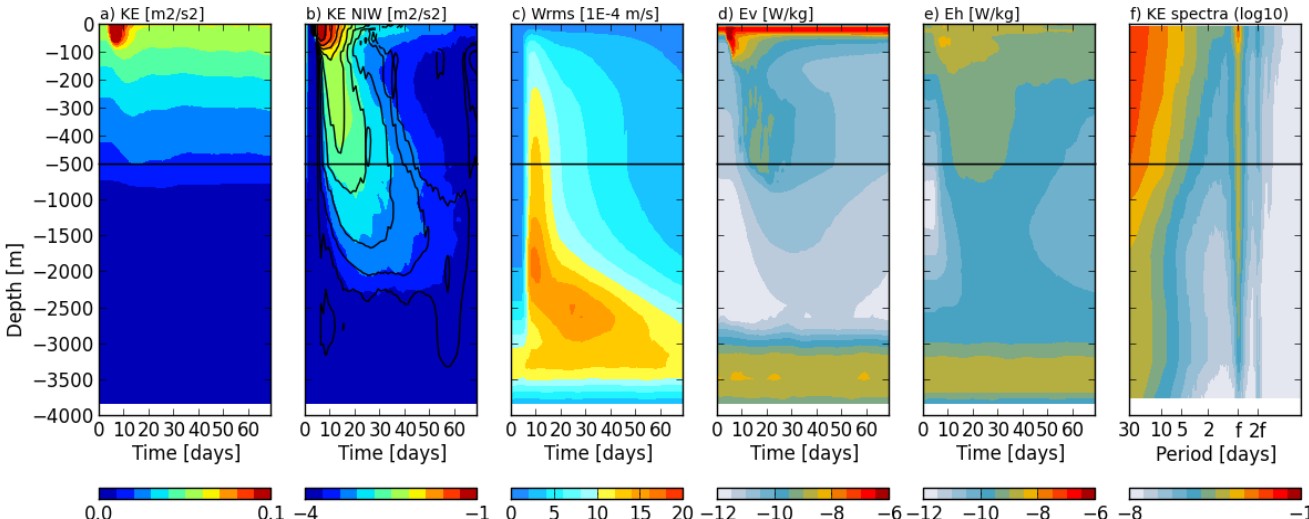

**Fig. 8.** Response of the ocean to the passage of a single storm : (a) horizontal kinetic energy (m$^2$ s$^{-2}$), (b) horizontal kinetic energy in the NIW band (colors, log10 m$^2$ s$^{-2}$) and difference of horizontal kinetic energy between the simulation with storms and a reference simulation without storms (iso-contours), (c) r.m.s. of the vertical velocity ($10^{-4}$ m s$^{-1}$) defined as $\sqrt{\langle w^2 \rangle}$ where ▨ is the horizontal average operator, (d) $\varepsilon_v$ energy dissipation due to vertical diffusion (W kg$^{-1}$) and (e) $\varepsilon_h$ the energy dissipation due to horizontal diffusion (W kg$^{-1}$). These diagnostics are spatially averaged between Ly/3 and 2Ly/3. The spatially averaged power spectra of the meridional velocity (log10 m$^2$ s$^{-2}$ day$^{-1}$) is shown in (f) and has been computed using hourly data from day 0 to day 70. The storm starts at day 3 and ends at day 7.




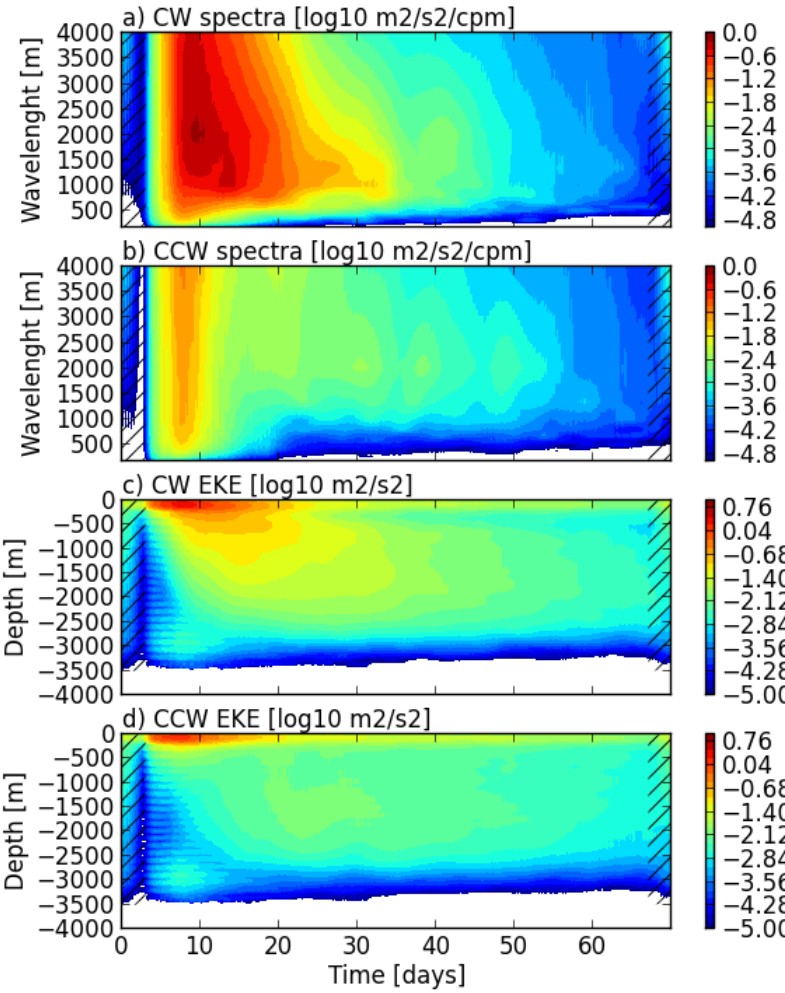

**Fig. 9** : Temporal evolution of clockwise (CW) and counterclockwise (CCW) spectra as a function of vertical wavelength, computed from WKB stretched near inertial velocities (a,b) for the single-storm experiment. Units are m$^2$ s$^{-2}$ cpm$^{-1}$. Near inertial KE computed as a function of time and depth from CW and CCW stretched velocities are shown in (c) and (d). Units are m$^2$ s$^{-2}$.



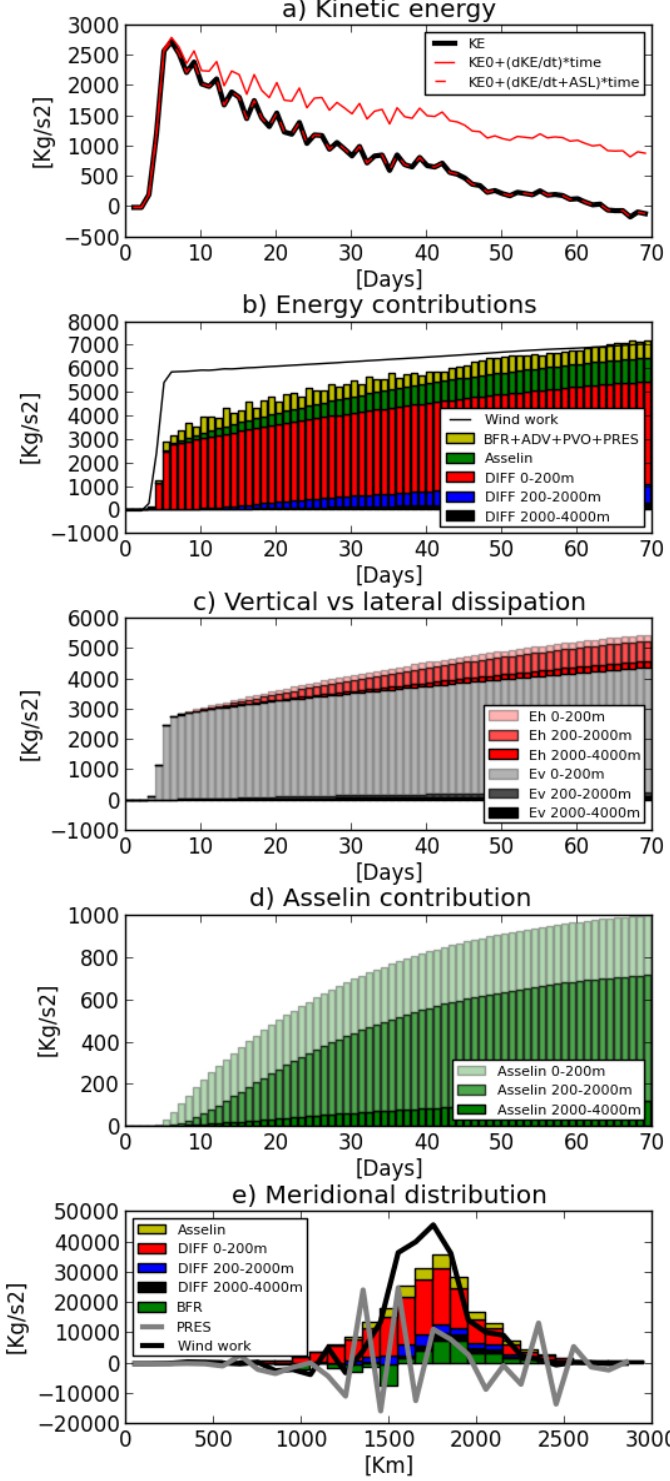

**Fig. 10.** Domain averaged response of the ocean to the passage of a storm from the same experiment already described in Fig. 5, 8 and 9. In order to isolate the response of the storm, we show here the differences with a reference experiment without storm and starting from exactly the same initial conditions. (a) Horizontal kinetic energy ($m^2 s^{-2}$) computed directly from model velocity (bold black) and indirectly from the time integral of kinetic energy tendency computed online before (red) and after (dotted red) Asselin time filtering. (b) Cumulated contribution of the different terms of the KE equation (DIFF represent the sum of both horizontal and vertical dissipations). (c) Cumulated lateral (Eh) and vertical (Ev) energy dissipation integrated in different depth ranges. (d) Cumulated dissipation of energy by the Asselin time filter integrated in different depth ranges. (e) Meridional distribution of cumulative wind work, viscous dissipation, bottom friction, horizontal pressure gradients, and Asselin energy dissipation at day 70.



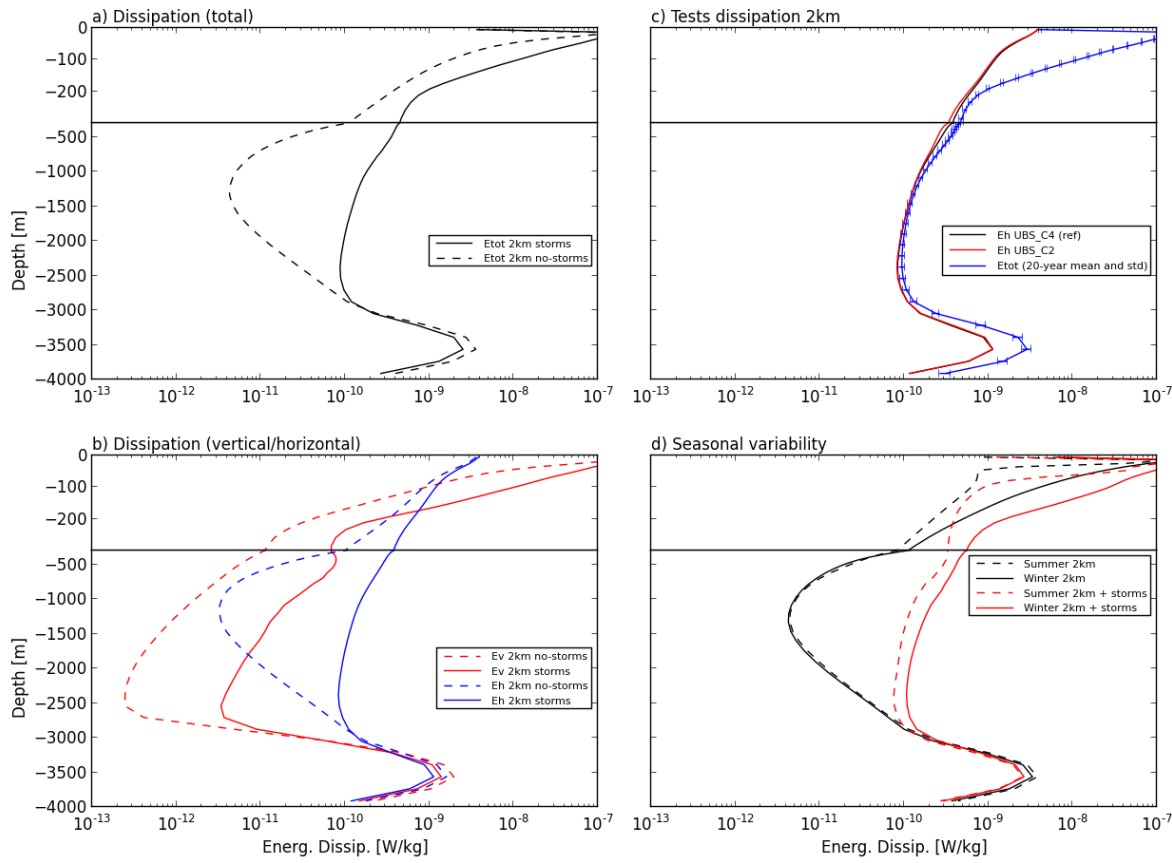

**Fig. 11.** Kinetic energy dissipation ($\varepsilon$; W kg$^{-1}$) as a function of depth in experiments at 2-km with storms (continuous lines) and without storms (dashed lines) : total energy dissipation $\varepsilon$ with and without storms (a), dissipation due to vertical processes $\varepsilon_v$ and dissipation due to horizontal processes $\varepsilon_h$ (b), $\varepsilon_h$ computed from a 2$^{nd}$ order (UBS_C2) or 4$^{th}$ order (UBS_C4) centered scheme (see text for details) together with a 20 years mean and standard deviation of $\varepsilon$ for the 2-km reference experiment (c), and summer (December, January, February) and winter (June-July-August) $\varepsilon$. Profiles are computed using 5-day snapshots of the entire domain for a 2 years period. Position, strength and duration of the storms remain strictly equal in the different experiments.



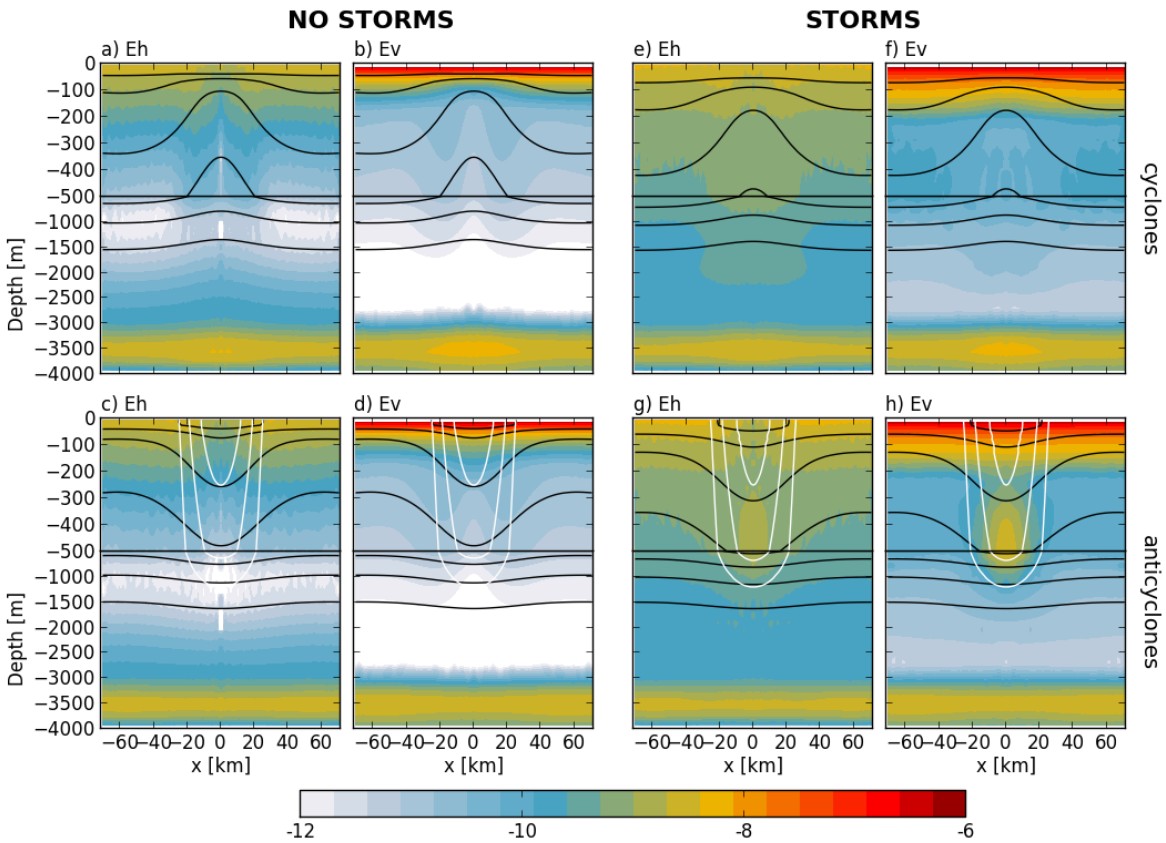

**Fig 12.** $\varepsilon_h$ and $\varepsilon_v$ (W kg$^{-1}$) distribution within composite cyclones (top) and anticyclones (bottom) identified in the 2-km experiments without storms (left) and with storms (right). The black iso-contours are isotherms from 2 to 8 °C and $\sigma/f$ iso-contours are shown in white (0.9, 0.95 and 0.98 $\sigma/f$), with $\sigma = f + \xi$ the effective frequency and $\zeta$ the relative vorticity. Composites are built using 10 years of 5-day averaged model outputs, between Ly/3 and 2Ly/3. A total of 8167 cyclone and 8878 anticyclone snapshots have been identified in the experiment without storms and 7306 cyclone and 8037 anticyclone snapshots in the experiment with storms.







**Fig. 13**. Kinetic energy dissipation ($\varepsilon$; W kg$^{-1}$) as a function of depth in experiments at 20-km, 5-km, 2-km and 1-km horizontal resolution, with storms (continuous lines) and without storms (dashed lines) : total energy dissipation $\varepsilon$ with storms (a) and without storms (b), dissipation due to vertical processes $\varepsilon_v$ with storms (c) and without storms (d), dissipation due to horizontal processes $\varepsilon_h$ with storms (e) and without storms (f), and the fraction of the total dissipation due to vertical processes ($\varepsilon_v / \varepsilon$ in %) (g and h). As in Figure 11, profiles are computed using 5-day snapshots of the entire domain for a 2 years period. Position, strength and duration of the storms remain strictly equal in the different experiments. The experiment z320 has an horizontal resolution of 2-km but 320 vertical levels, ranging from 1 meter at the surface to 250 meters at the bottom (below 2500 m depth the vertical size of the cells is the same as in the 2-km reference experiment).



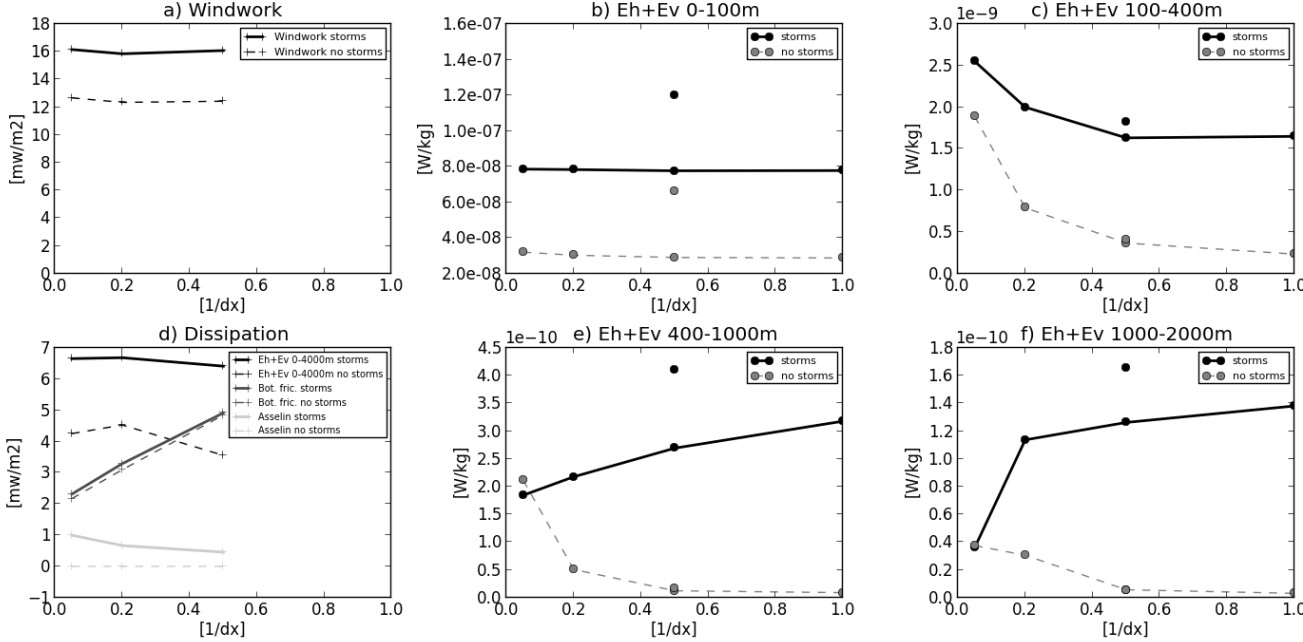

**Fig. 14.** Kinetic energy dissipation (ε) and wind work as a function of model resolution, in experiments with (continuous lines) and without storms (dashed lines) : (a) wind work and energy dissipation integrated from surface to bottom (mW m$^{-2}$), (d) energy dissipation integrated from surface to bottom (decomposed into contributions from ε, bottom friction and Asselin time filter; mW m$^{-2}$) and total dissipation ε (W kg$^{-1}$) averaged in the depth ranges 0-100m (b), 100-400m (c), 400-1000m (d) and 1000-2000m (e). Values are computed using 5-day snapshots of the entire domain for a 2 years period as in Fig. 9. Isolated dots represent ε for the 2-km experiment with 320 vertical levels. Wind work (a) and energy dissipation contributions (d) have only been computed for the 20, 5 and 2-km experiments.




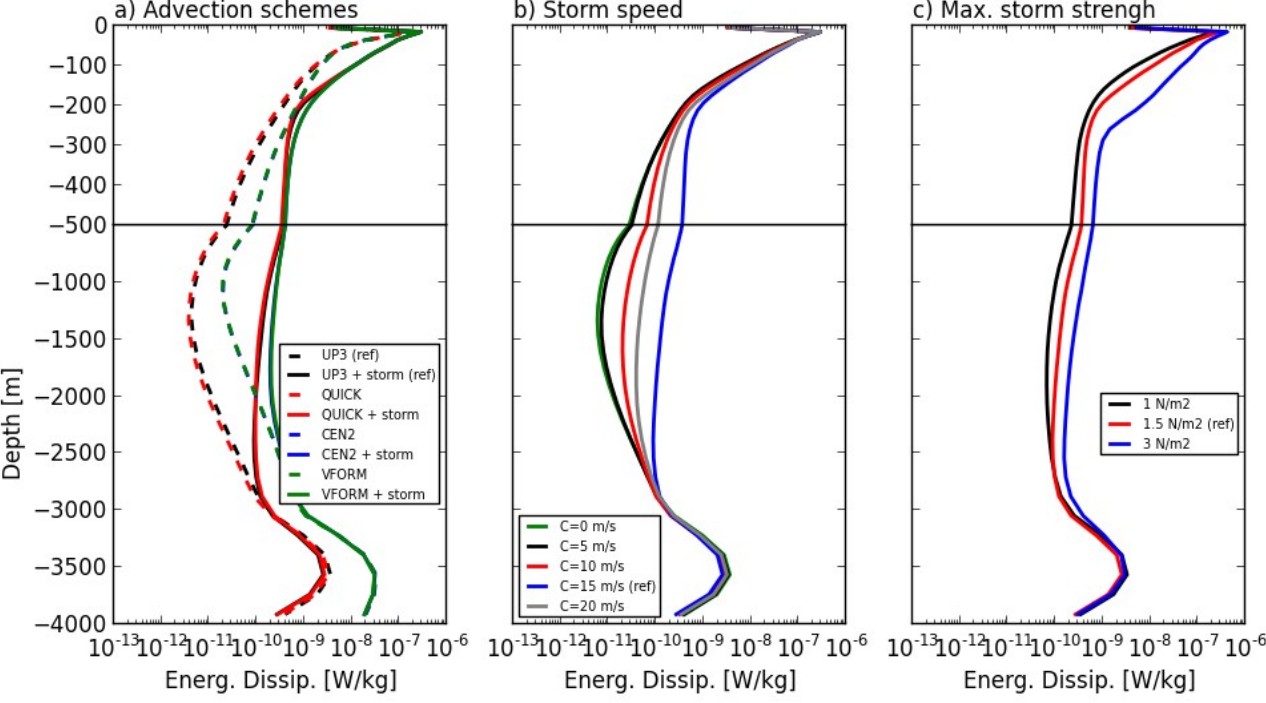

**Fig. 15.** Sensitivity of energy dissipation (ε) profiles to numerics (a), storm speed (b) and storm strength (c). Experiments with (without) storms are shown with continuous (dashed) lines. The advective schemes tested in (a) are UP3 (reference), QUICK, flux form 2nd order centered advection scheme, and a vector form advection scheme. The profiles of the latter two (blue and green colors) are confounded in panel (a). Dissipation induced by storms traveling at different speeds is tested in (c) for propagation speeds of 0, 5, 10, 15 and 20 m s$^{-1}$. In these experiments the duration and the power of the storms are the same as in the reference experiment (for which the storm propagation speed is 15 m s$^{-1}$). In (d), the sensitivity to the storm strength is tested by comparing experiments with maximum wind stress values equal to 1, 1.5 (reference) and 3 N m$^{-2}$. All the sensitivity experiments are run at 2-km horizontal resolution. They start from the same initial condition equilibrated without storms, and they are run for three years. Profile are built using 5-day snapshots of the entire domain for the last two years of the simulations.



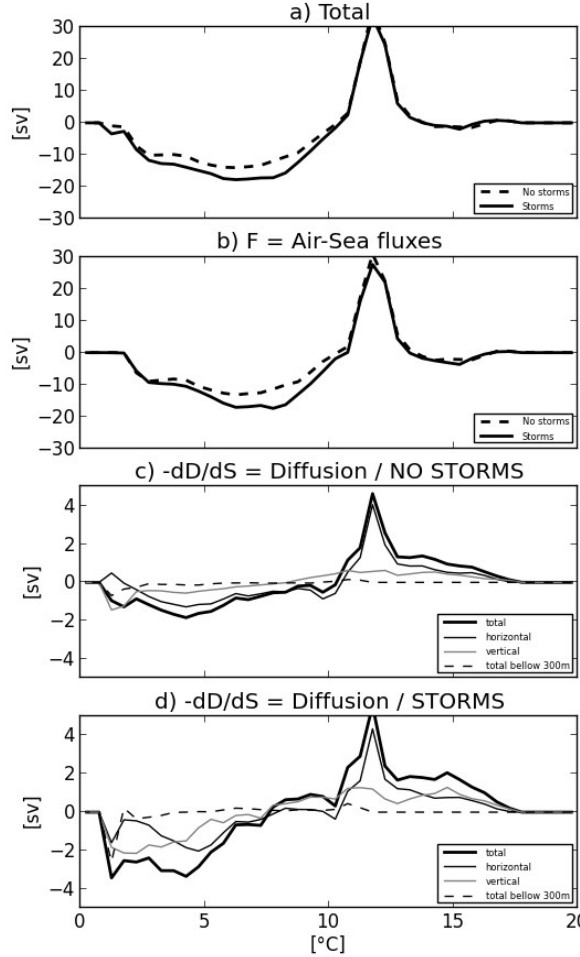

**Fig. 16.** Transformation rate (in Sv) : total (a), contribution of air-sea fluxes (b) and diffuse fluxes across isotherms for the 2-km simulations without storms (c) and with storms (d). The diffuse fluxes are separated into vertical (light gray) and lateral (black) contributions. The dashed lines in (c) and (d) correspond to transformation by diffuses fluxes below 300 m depth. Model transports have been multiplied by 10 in order to scale them to the full Southern Ocean.