# Peer review of "Dissipation of the energy imparted by mid-latitude storms in the Southern Ocean"

_Ocean Science, 2016_

## Referee Comment (RC1) · Anonymous Referee #1 · 17 Feb 2016

This manuscript addresses the fate of energy (mostly near-inertial energy) imparted by Southern Ocean storms and its effect on the Southern Ocean overturning circulation using a suite of semi-idealized high-resolution ocean models. The numerical experiments designed by the authors are suitable to address this important topic. Although the results on near-inertial energy generation, propagation and dissipation are somewhat expected, its effect on Southern Ocean overturning circulation is new. Furthermore, I haven't seen sensitivity studies of near-inertial energy budget to model numerics at such high resolutions before. Therefore I think this manuscript will make a useful and necessary contribution to this topic. I have some comments for the authors to consider, but most of my comments are minor in nature.

1. The manuscript talks about "energy imparted by mid-latitude storms". Well, it is not strictly correct. The synoptic winds associated with mid-latitude storms not only

generate near-inertial waves via time-varying wind stress, but also contribute significantly to the time-mean wind stress at mid-latitudes via the nonlinear dependence of wind stress on the wind. For example, Zhai et al. (JPO, 2012) showed that when the synoptic winds are included in the stress calculation, the wind power input to the ocean general circulation can increase by as much as 70%.

2. Recently, Rath et al. (JGR, 2013) found that accounting for the ocean-surface-velocity dependence of the wind stress leads to a large reduction of wind-induced near-inertial energy of approximately 40% in their 1/10 degree Southern Ocean model. This relative wind damping effect seems to be a very important way of dissipating near-inertial energy in the ocean. When you force your model directly with wind stress that knows nothing about the surface ocean currents, this relative wind damping effect is absent. I think this issue should at least be discussed, given that this manuscript is about energy dissipation.

3. line 19-20 on page 3. Should be "2000 km long" and "3000 km wide".

4. line 26 on page 3 and in many other places in the manuscript. It should be "3x104".

5. The last sentence on page 3 needs to be rephrased. It is not clear to me what you want to say here.

6. In the model configuration section on page 4, there is no information about the existence of the background wind forcing without storms at all. It should not be left completely in the appendix.

7. line 21 on page 4. Brackets are needed for "2012".

8. page 6. (2) also includes diffusive energy transport in the vertical direction, not just energy dissipation?

9. line 12 on page 7. The average KE exceeds......, but the caption of Fig. 6a says "EKE"?

10. line 1 on page 9. What is WKB stretched CW and CCW? Need to explain it or at least point the readers to the appendix.

11. line 21-22 on page 9. Why does the storm lead to strengthening of the eastward current?

12. line 25 on page 9. The Coriolis force is perpendicular to the current, and therefore should not do any work?

13. line 17 on page 10. The authors should check what these papers say before citing them here.

14. line 21 on page 10. You need to explain how the strength of the storm activity is varied seasonally here, or at least point the readers to the appendix.

15. line 31 on page 10. Again, why does the mean KE increase in response to the storms?

16. line 7 on page 11. How do you define the mixed layer?

17. line 23 on page 13. The effective frequency should be "$f+\zeta/2$", according to Kunze (JPO, 1985)?

18. line 12 on page 17. Should be "easy comparison".

19. line 22 on page 17. I assume that changes in air-sea fluxes are due to the feedback term in surface heat flux forcing? If so, need to say it.

20. lines 13-14 on page 19. It is not clear to me why you conclude that your results are noticeably different from the previous two studies, since your results also show that the majority of energy imparted by the storms is dissipated within the top 200 m.

21. line 18 on page 19. What is that part of wind work that is not near-inertial wind work? Wind energy input to the surface Ekman currents?

22. Figure 5. Three different color maps are used. Is this necessary? Why is 17 days

after the passage of the storm is chosen? Is it to give enough time for the near-inertial waves to reach the base of the anticyclonic eddy?

23. Figure 8. a) and b) are plotted differently (one on log scale and one not), which makes it hard to compare them quantitatively. In the caption for c), the symbol "< >" appears messed up on my printed-out version.

---

## Referee Comment (RC2) · Anonymous Referee #2 · 18 Feb 2016

The authors set out to examine the dissipation of near inertial energy imparted by mid-latitude storms in the Southern Ocean. Idealized experiments are designed and utilized that seem well suited for this examination. The analysis is well designed and leads to conclusions on the generation and fate of near inertial energy that are not physically surprising. The influence of Southern Ocean storms on the MOC in models is certainly new. Further, the influence of model parameters is novel and important for other models of the Southern Ocean to consider. The work in this manuscript is well thought out and will make an excellent addition to the Southern Ocean literature. My comments, for the most part are minor, but I offer a few thoughts and possible additions for the authors.

The influence of Southern Ocean storms on MOC are certainly interesting. They seem to be in partial agreement with the observational results of Hogg et al. (JGR, 2015) who

examined the response of EKE and MOC to increases in southern ocean winds. Given that Figure 7 represents a 10 year average, the average storm influence is behaving a bit like a somewhat uniform increase in winds. In fact, I wonder if you may find similar MOC results for an experiment where the wind stress was uniformly increased everywhere since the storm tracks may average out over the 10 year cycle to a *roughly* uniform increase in the domain. Either way, it seems that your results are consistent and you might consider referencing Hogg et al. On the other side, a recent paper by Gent (2016, Ann. Reviews) suggests that southern ocean winds cannot be a primary driver of MOC due to eddy compensation. I think placing your results in the context of these two types of results would be useful.

Expanding on a comment from the other referee, I think the details of your momentum transfer parameterization are important. In addition to inclusion of the moving ocean, I think the role of wave generation is important. Some of the momentum input due to storms will go into wave generation and not directly into the local currents. The waves may break far from the cyclone thus changing the local dissipation profiles (see for example Suzuki et al 2014, JPO or Curcic 2015, UMiami Dissertation). Further, wave mean flow interactions (e.g. langmuir cells) may dramatically modify local dissipation. This result may be difficult to tease out as the wave drift relative to currents will rotate 180 degrees as the storm passes over (e.g. Sullivan et al 2012, JPO). No further experiments are necessary, but a short discussion on possible effects would be interesting.

In a few places in the manuscript the submesoscales are referenced. For example, section 3 seems to suggest that the submesoscales are for $\lambda < 60km$. This seems incredibly large. The first rossby radius of deformation computed from a model configuration not unlike that presented here (computed via the method in Chelton et al 1998, JPO) is ~10km near the ACC implying that the submesoscales are even smaller. Using scalings from Fox-Kemper et al (2008, JPO) submesoscales in the ACC are roughly 2 km, suggesting your 1km results are borderline submesoscale permitting. This suggests that the shallowing of the mixed layer noted in section 5 is most likely not due to the submesoscales.

In your paper you note the importance of resolving mixed layer dynamics (pg 15), yet you never discuss or test the sensitivity of your results to different mixed layer parameterizations. I would expect that accurate representation of mixed layer processes will have an impact on your results. For example, the GLS scheme could be used with different parameters to produce $\kappa - \omega$ or $\kappa - l$ schemes. Further, to the best of my knowledge the GLS scheme does not consider non-local processes. While I agree with your assessment that this will not matter significantly during the storms passage, I would point out that the non-locality (most active from the prior winter) may change the depth of the mixed layer prior to storm passage, which in turn may influence your results of NIE fate. While KPP certainly has its own issues, the inclusion of non-locality may be important. Why did you choose not to consider other vertical mixing parameterizations except the GLS $\kappa - \epsilon$ form?

**Specific Comments**

**Page 1**

–Line 28 – efforts should be effort

–Line 29 – with should be to, and semi-colon needed after on

**Page 4**

–Line 13 – NEMO is hydrostatic, so it really doesn't solve the three-dimensional primitive equations, correct? I would assume w is diagnosed from integrated divergence. I think you could just leave out that phrase and say it is discretized on...

– Line 18 – you reference the resolution of layers in the vertical for the 320 m test, but never for the 50 m baseline. It would be nice for a comparison

**Page 5**

–Line 19 – is there any sensitivity to the chosen 70-days (during the final year of the simulation) over which you average?

**Page 6**

–Line 20 – again, how are you determining that $\lambda < 60km$ is submesoscale?

**Page 7**

– Paragraph 1 – What is the vertical coordinate treatment in NEMO relative to the bottom? Do you use terrain following / z with partial bottom cells / z with shaved cells / something else? This feels like an important point as it will have an impact on how the flow interacts with the bottom topography in the model.

– line 8 / 9 – starting at the absence of topographic, I think it should be something like "the absence of a topographic ridge and narrow passages does not allow us to obtain"

**Page 10**

– line 2 – what is your near bottom resolution in the 50 layer, baseline case? Could this explain the absence of enhanced bottom dissipation as you probably are not resolving the bottom boundary layer?

**Page 11**

– line 8 – no comma is needed after fig 6e

– line 14 – should be – the submesoscale

**Page 12**

– line 2 – the sentence beginning with – At the difference of this general balance – is very confusing to me, in particular the first two clauses. It also feels like this could be broken into multiple sentences, perhaps after the second comma.

– line 32 – it feels like coherent should be consistent

**Page 13**

– paragraph 1 of section 5.3 – does your analysis here have any dependence on the assumed diameter and or the threshold deviation from a circular shape? For example, in your vorticity snapshot, many of the eddies are very elongated.

**Page 14**

– line 29 - 30 (and other places) (resp. ....) I am unfamiliar with this notation. I think this is 'respectively', but am unsure. If it is, I don't think that is necessary. Most often the respectively is omitted.

**Page 15**

–Vertical resolution section, did you consider sensitivity to resolution at the mixed layer interface? Appropriate simulation of entrainment/detrainment should have non-negligible impacts on your results. Your 50 vs 320 layer case may have sufficient enough differences in the respective resolutions at the boundary layer depth to speak to this already.

–Advection schemes, what resolution was the biharmonic viscosity coefficient used with and was it scaled with resolution? Was it actually used in other simulations? If it was used and not varied with resolution, it seems like your high-resolution simulations would be overly diffuse.

–Advection schemes, Note that when you compute numerical diffusion relative to a even order scheme, it is a slight over-estimate as these schemes, $2^{nd}$ order especially are anti-diffusive. It would be nice to make a quick mention of this in the text.

**Page 18**

– line 18 – parenthesis needed around 2011

**Page 21**

– line 11 – The sentence beginning Although the settings have... is very confusing to me. I think it might be fixed by adding a comma after "differences" and change "consists in" to is

**Page 22**

– line 24 – Life time should be The lifetime

– line 26 – affect should be affects

– line 26 – each should be every

**Figure 2**

–Can you explain why the MLD bias is so large in what I believe is the restoring region $(y > 2600km)$? It seems that if you restore to something resembling observations you shouldn't have such a large bias. Or is this related to the comparison to de Boyer Montegut versus some other climatology?

–I would move the clause mentioning how the MLD is calculated up in the description to where the MLD is introduced (first line)

**Figure 6**

Is the MLD defined as in Figure 2? it should be stated.

**Figure 7**

– The subscripts on the plot labels $\Psi$ in (c) - (f) are very difficult to see, but I think they say "iso"? Could these be enlarged?

– do the dotted and dashed lines in (e) - (f) and (c) - (d) respectively represent the same thing (likelihood of a parcel being at the surface)? The caption only mentions dotted lines.

– I think you should remove the 'and' following Abernathy et al 2011 and add a comma

– Do the dotted lines in (e)/(f) correspond at all to modeled mixed layer depths? It would be interesting to see if all buoyancy classes that don't exist at least part of the time always reside in the mixed layer or not.

**Figure 8**

– be consistent between the caption and panel for (d) and (e) you have Ev Eh versus $\epsilon_v$ and $\epsilon_h$

– the 10 after log10 should be subscript

– There is an odd symbol after your rms definition, I'm guessing it is <>

**Figure 10**

– Why did you choose to separate your analysis at 200 m depth? Is this a rough estimate of the boundary layer depth? Or some change in the shape profile perhaps?

**Figure 11**

– Should "UBS_C2" be "UBS - C2" instead? Same for "UBS_C4"?

**Figure 13**

– If the horizontal dissipation is due to advective scheme biases, it seems that this dissipation should reduce (assuming it is the implicit diffusion due to upstream bias) and not increase in panel h. I can't seem to find this panel discussed in text, but may have missed it.

**Figure 15**

– the abbreviation VFORM should be included after "vector form" in the caption.

---

## Editor Comment (EC1) · M. Hecht (Editor) · 14 Apr 2016

Dear Authors, I can understand that the question of "Why did you choose not to consider other vertical mixing parameterizations..." would be a more difficult one to fully accommodate. If you will acknowledge the influence of unresolved mixed layer processes, mention why GLS was chosen and perhaps also acknowledge that good results at ocean station P do not necessarily guarantee good results in the Southern Ocean, then I believe that Referee 2 will be satisfied that this question has been adequately addressed.

Sincerely yours, –Matthew Hecht

---

## Author Comment (AC2) · 14 Apr 2016

The authors set out to examine the dissipation of near inertial energy imparted by mid-latitude storms in the Southern Ocean. Idealized experiments are designed and utilized that seem well suited for this examination. The analysis is well designed and leads to conclusions on the generation and fate of near inertial energy that are not physically surprising. The influence of Southern Ocean storms on the MOC in models is certainly new. Further, the influence of model parameters is novel and important for other models of the Southern Ocean to consider. The work in this manuscript is well thought out and will make an excellent addition to the Southern Ocean literature. My comments, for the most part are minor, but I offer a few thoughts and possible additions for the authors.

The influence of Southern Ocean storms on MOC are certainly interesting. They seem to be in partial agreement with the observational results of Hogg et al. (JGR, 2015) who examined the response of EKE and MOC to increases in southern ocean winds. Given that Figure 7 represents a 10 year average, the average storm influence is behaving a bit like a somewhat uniform increase in winds. In fact, I wonder if you may find similar MOC results for an experiment where the wind stress was uniformly increased everywhere since the storm tracks may average out over the 10 year cycle to a roughly uniform increase in the domain. Either way, it seems that your results are consistent and you might consider referencing Hogg et al. On the other side, a recent paper by Gent (2016, Ann. Reviews) suggests that southern ocean winds cannot be a primary driver of MOC due to eddy compensation. I think placing your results in the context of these two types of results would be useful.

By construction the 10 year average wind stress is the same in the storms and no-storms simulations, so our model setup with storms does not include the response of the ocean to modifications of the timemean wind stress by the storms. This was a deliberate choice so we were sure that the sensitivity of the solution was not due at first order to changes in the time-mean wind stress.

We agree that it would be interesting to understand whether the average storm influence on the MOC is behaving like an uniform increase in winds. A new set of simulations would be required to properly answer this question. But part of the answer is already given in the manuscript :

"Although the settings have differences, an instructive comparison is estimating the change in mean wind stress required to increase the upper MOC cell (the only one we simulate) by 16% in the sensitivity experiments carried out by Abernathey et al (2011). Their figure 5 indicates a change from 0.20 N m-2 to ~0.23 N m-2 (+15%) is needed when interactive air-sea flux are used. This further confirms the importance of synoptic winds."

If we would not remove the residual mean signature of the storms, the mean wind profile of our simulation would be as follows (the inclusion of storms lead to a 1.3% decrease of the domain average wind stress):

So the residual wind modification is much less than the +15% increase estimated from the results by Abernathey et al. (2011). This indicates that a simulation forced with this uniform residual wind shall

not increase the MOC as in the experiments with storms. We prefer not to discuss this point in the manuscript since we do not provide a thorough demonstration. Indeed, it would required some confirmation by running and analyzing additional experiments (our access to HPC resources is closed).

Thanks for the reference to Hogg et al. (2015) and Gent (2016). They have been added in the conclusion.

Expanding on a comment from the other referee, I think the details of your momentum transfer parameterization are important. In addition to inclusion of the moving ocean, I think the role of wave generation is important. Some of the momentum input due to storms will go into wave generation and not directly into the local currents. The waves may break far from the cyclone thus changing the local dissipation profiles (see for example Suzuki et al 2014, JPO or Curcic 2015, UMiami Dissertation). Further, wave mean flow interactions (e.g. langmuir cells) may dramatically modify local dissipation. This result may be difficult to tease out as the wave drift relative to currents will rotate 180 degrees as the storm passes over (e.g. Sullivan et al 2012, JPO). No further experiments are necessary, but a short discussion on possible effects would be interesting.

We would prefer not to risk us to discuss on possible effects of these different processes; it would be too speculative. Nevertheless we add a general paragraph in the discussion Section 7.2 Energy pathways that point to these shortcomings:

"The vertical turbulence model we use does not include an explicit wave description so the surface wave mixing effect is parameterized and non-local wave breaking, Stokes drift or Langmuir cells are not considered. These processes modulate the momentum and energy deposited into the ocean as well as near surface dissipation rates. For example, the analysis of a coupled atmosphere-wave-ocean model simulating hurricane conditions suggests that the Stokes drift below the storm can contribute up to 20% to the Lagrangian flow magnitude and change its orientation (Curcic et al. 2016). These processes certainly impact the near-inertial wind energy input and distribution of its dissipation, and would deserve further attention, perhaps using a more realistic (regional) setup "

In a few places in the manuscript the submesoscales are referenced. For example, section 3 seems to suggest that the submesoscales are for < 60km. This seems incredibly large. The first rossby radius of deformation computed from a model configuration not unlike that presented here (computed via the method in Chelton et al 1998, JPO) is 10km near the ACC implying that the submesoscales are even smaller. Using scalings from Fox-Kemper et al (2008, JPO) submesoscales in the ACC are roughly 2 km, suggesting your 1km results are borderline submesoscale permitting. This suggests that the shallowing of the mixed layer noted in section 5 is most likely not due to the submesoscales.

We guess, that the confusion comes from that in our manuscript, we refer to the submesoscale range by considering the wavelengths ( $2\pi/k < 60$ km, with k the wavenumber) and not the length scale (1/k < 10km). Note that Fox-Kemper et al (2008, JPO) used a model 2-km resolution to study sub-mesoscale re-stratification effects.

When introducing the sub-mesoscale range, we now precise : "(i.e. horizontal scale below 10km)".

In your paper you note the importance of resolving mixed layer dynamics (pg 15), yet you never discuss or test the sensitivity of your results to different mixed layer parameterizations. I would expect that accurate representation of mixed layer processes will have an impact on your results. For example, the GLS scheme could be used with different parameters to produce k-w or k-l schemes. Further, to the best of my knowledge the GLS scheme does not consider non-local processes. While I agree with your assessment that this will not matter significantly during the storms passage, I would point out that the non-locality (most active from the prior winter) may change the depth of the mixed layer prior to storm passage, which in turn may influence your results of NIE fate. While KPP certainly has its own issues, the inclusion of non-locality may be important. Why did you choose not to consider other vertical mixing parameterizations except the GLS k- $\epsilon$  form ?

Indeed, we leaved aside the sensitivity of our solutions to the vertical subgrid scale physics. We did this choice first because of a limited access to HPC resources so we were forced to prioritize our sensitivity experiments. The range of numerical and physical choices that affect the vertical physics is very large (closure type, stability function, inclusion of non local effect due to static instabilities, Langmuir Cell effect, surface roughness computation, varying wave field, etc...) so long term, 3D, high-resolution experiments may not be well suited to perform exhaustive series of sensitivity experiments that could benefit to the community. So we choose the vertical mixing scheme setting available in NEMO that gave the most satisfactory results at high latitudes (it has been tuned at station PAPA, 50N by Reffray et al. 2015).

Specific Comments

-Line 28 - efforts should be effort

Corrected.

-Line 29 - with should be to, and semi-colon needed after on

Corrected.

-Line 13 – NEMO is hydrostatic, so it really doesn't solve the three-dimensional primitive equations, correct? I would assume w is diagnosed from integrated divergence. I think you could just leave out that phrase and say it is discretized on...

*Yes, w is diagnosed from integrated divergence. So we removed "three-dimensional", but we retained that the model solves the primitive equations.*

- Line 18 - you reference the resolution of layers in the vertical for the 320 m test, but never for the 50 m baseline. It would be nice for a comparison

The resolution of layers was already given for both 50-levels and 320-levels configurations, in the model description configure (Section 2.1) and in the description of the sensitivity tests (Section 5.4), respectively.

-Line 19 – is there any sensitivity to the chosen 70-days (during the final year of the simulation) over which you average?

The one-storm simulation has been performed starting with initial conditions typical of the austral summer. We did not performed sensitivity tests by changing the initial conditions. The aim of the single-storm experiment is mainly to offer a general/qualitative view of the dynamics and main energy balance involved in the response of the ocean to the passage of one storm. Since the results shown in this section are in full agreement with the analysis of the "multi-storm" experiments in the subsequent sections, we have not felt necessary to perform additional experiments.

-Line 20 – again, how are you determining that < 60km is submesoscale?

Again, this depends on whether we consider wavelength (<60km) or horizontal scale (<10km).

**Page 7**

- Paragraph 1 - What is the vertical coordinate treatment in NEMO relative to the bottom? Do you use terrain following / z with partial bottom cells / z with shaved cells / something else? This feels like an important point as it will have an impact on how the flow interacts with the bottom topography in the model.

We use partial steps and it is now mentioned in the description of the model.

- line 8 / 9 - starting at the absence of topographic, I think it should be something like "the absence of a topographic ridge and narrow passages does not allow us to obtain"

Corrected.

- line 2 – what is your near bottom resolution in the 50 layer, baseline case? Could this explain the absence of enhanced bottom dissipation as you probably are not resolving the bottom boundary layer ?

Bottom resolution in both 50 and 320 level configurations is 175m (this is mentioned in the model configuration section), so the bottom boundary layer (if we consider this layer as the near field where internal waves generated at the bottom break) is not well resolved. We think that the main reason that explain the absence of enhanced bottom dissipation is the one given in the manuscript : "The levels of near inertial energy below 2500 m depth remain 2 to 3 order of magnitude lower than those found in the mixed-layer and are not sufficient to significantly increase bottom dissipation."

- line 8 - no comma is needed after fig 6e

Corrected.

- line 14 - should be - the submesoscale

Corrected.

- line 2 - the sentence beginning with - At the difference of this general balance - is very confusing to me, in particular the first two clauses. It also feels like this could be broken into multiple sentences, perhaps after the second comma.

The sentence has been rephrased so we hope it is now less confusing : "The KE balance in both experiments are very similar, with overall wind work mainly balanced by the work done by bottom friction (38.9% without storms and 30.5% with storms), pressure work maintaining the system available potential energy (32.2%, 26.0%) and vertical diffusion (23.4%, 33.1%). At the difference of this general balance, and in agreement with results for the single-storm experiment described in section 4, t. The KE balance also indicates that the additional input of energy provided by the storms (+3.64 mW m-2) is balanced at 90% by dissipation (-2.86 mW m-2 for horizontal and vertical dissipation to which one should add the Asselin filter contribution) with pressure work and bottom friction being secondary (respectively -0.18 mW m-2 representing a 5% contribution and -0.07 mW m-2 representing a 2% contribution). This is in stark contrast with the equilibration of the low-frequency wind work feeding the balanced circulation."

- line 32 - it feels like coherent should be consistent

This has been corrected.

- paragraph 1 of section 5.3 – does your analysis here have any dependence on the assumed diameter and or the threshold deviation from a circular shape? For example, in your vorticity snapshot, many of the eddies are very elongated.

The cyclone/anticyclone composites tend to resemble more and more when decreasing the size of the selected eddies or when being less restrictive on the shape of the eddies. On the contrary a more restrictive criteria on both shape and size leads to more contrasted results between cyclones and anticyclones but turns representative of only a very small fraction of the domain. So we choose a compromise between both in order 1) to have a total amount of dissipation contained in the composite that would be significant for the model domain and 2) to be enough selective so compositing allows to show contrasted dissipation profiles between cyclones and anticyclones.

- line 29 - 30 (and other places) (resp. ....) I am unfamiliar with this notation. I think this is 'respectively', but am unsure. If it is, I don't think that is necessary. Most often the respectively is omitted.

We removed the use of resp. (omitting them or rephrasing) at the exception of page 18 where we think this notation helps the reader.

-Vertical resolution section, did you consider sensitivity to resolution at the mixed layer interface? Appropriate simulation of entrainment/detrainment should have nonnegligible impacts on your results.

Your 50 vs 320 layer case may have sufficient enough differences in the respective resolutions at the boundary layer depth to speak to this already.

In the 320 layer case, the resolution increase mainly affect the upper ocean (2m resolution at the surface instead of 10m in the 50 level configuration) while the size of the cells below 2500 meters are equal to the 50 level experiment so that the local characteristics of flow-topography interactions are unchanged.

Dissipation is altered in the upper levels (see Fig. 14b), but we cannot conclude whether the differences in terms of dissipation in the interior (from 150 m to 2500meters) are due to sensitivity to resolution at the ML interface or modification of the NIW propagation well below the mixed layer. So we prefer not to conclude on the reasons for the modifications in the interior, and keep the comment as follows :

"The overall dissipation  $\varepsilon$  is increased in presence of storms in the interior in the configuration with 320 vertical levels (Fig. 13a,b and Fig. 14c-e), indicating that the downward propagation of the NIE is better resolved in the high vertical resolution experiment with more NIE available at depth. Similar increase of  $\varepsilon$  in the upper 100m in the experiments with and without storms (Fig. 14b) suggests that mixed-layer dynamics are profoundly altered when changing the vertical resolution. "

-Advection schemes, what resolution was the biharmonic viscosity coefficient used with and was it scaled with resolution? Was it actually used in other simulations? If it was used and not varied with resolution, it seems like your high-resolution simulations would be overly diffuse.

The explicit biharmonic diffusion has been added to the experiments with 2-km horizontal resolution, that use 2nd order advection scheme in flux form (CEN2) or vector invariant form (VFORM): 2km-nostorm\_CEN2, 2km-storms\_CEN2, 2km-notorm\_VFORM, 2km-storms\_VFORM (see Table 1). We used the same biharmonic viscosity coefficient than Levy et al. (Ocean Modelling, 2012) for their 2-km simulation. The 1-km simulations have been performed here with UP3 scheme only, so diffusion was implicit.

-Advection schemes, Note that when you compute numerical diffusion relative to a even order scheme, it is a slight over-estimate as these schemes, 2nd order especially are anti-diffusive. It would be nice to make a quick mention of this in the text.

It is not completely clear for us whether we over-estimate or underestimate the dissipation (in particular for the 4th order scheme which is not strictly a 4th order scheme). So we prefer to keep the following sentence : "The 4th order scheme in NEMO involves a 4th order interpolation for the evaluation of advective fluxes but their divergence is kept at 2nd order, making the scheme not strictly non-diffusive. Although the estimation of UP3 horizontal diffusion depends on the scheme used as a reference we verify in section 5 that the sensitivity of domain averaged  $\varepsilon_h$  to the choice of the 2nd or 4th order scheme is much smaller than that resulting from other parameter changes, e.g., small changes in

the characteristics of the atmospheric forcing."

- line 18 - parenthesis needed around 2011

Corrected.

- line 11 – The sentence beginning Although the settings have... is very confusing to me. I think it might be fixed by adding a comma after "differences" and change "consists in" to is

Corrected following your suggestion.

- line 24 – Life time should be The lifetime

Corrected.

- line 26 - affect should be affects

Corrected.

- line 26 - each should be every

Corrected.

Figure 2

-Can you explain why the MLD bias is so large in what I believe is the restoring region (y > 2600km)? It seems that if you restore to something resembling observations you shouldn't have such a large bias. Or is this related to the comparison to de Boyer Montegut versus some other climatology?

The restoring is only for y > 2850km and is quite weak up to 2950 km, so the MLD bias occurring north of y=2500km is only partly corrected (from y=2850 where it tends to decrease). Achieving a realistic temperature and mixed-layer depth seasonal cycle has been very challenging. In particular the use of a temperature only equation of state complicated the problem.

-I would move the clause mentioning how the MLD is calculated up in the description to where the MLD is introduced (first line)

Done.

Figure 6

Is the MLD defined as in Figure 2? it should be stated.

Yes. It is now stated in the caption of Figure 6.

Figure 7

- The subscripts on the plot labels in (c) - (f) are very difficult to see, but I think they say "iso"? Could these be enlarged?

The titles of figure 7 have been enlarged.

- do the dotted and dashed lines in (e) - (f) and (c) - (d) respectively represent the same thing (likelihood of a parcel being at the surface)? The caption only mentions dotted lines.

In (c,d) the dashed lines corresponds to the cumulative distribution of SST while in (e.f) they correspond to the cumulative distribution of the MLD. This was not explicit in the caption and now it has been improved.

- I think you should remove the 'and' following Abernathy et al 2011 and add a comma

This has been corrected following your suggestion.

- Do the dotted lines in (e)/(f) correspond at all to modeled mixed layer depths? It would be interesting to see if all buoyancy classes that don't exist at least part of the time always reside in the mixed layer or not.

Yes they represent the average depth of the 10%, 50% and 90% deepest ML, illustrating that transformations occurs in buoyancy classes well below the mixed-layer, probably achieved by the shear of the eddies in the case without storms (e.g. see Figure 5d,e) and reinforced by the shear of the NIW when storms are included (Figure 5i,h)

**Figure 8**

- be consistent between the caption and panel for (d) and (e) you have Ev Eh versus v and h

The titles of panel (d) and (e) have been corrected.

- the 10 after log10 should be subscript

**Corrected.**

– There is an odd symbol after your rms definition, I'm guessing it is <> Figure 10

**Corrected.**

- Why did you choose to separate your analysis at 200 m depth? Is this a rough estimate of the boundary layer depth? Or some change in the shape profile perhaps?

We choose to separate our analysis at 200m depth because the average dissipation profile (e.g. Figure 11a) shows an inflexion at this depth suggesting a change of regime down to this depth.

Figure 11

- Should "UBS\_C2" be "UBS - C2" instead? Same for "UBS\_C4"?

This has been corrected.

Figure 13

- If the horizontal dissipation is due to advective scheme biases, it seems that this dissipation should reduce (assuming it is the implicit diffusion due to upstream bias) and not increase in panel h. I can't seem to find this panel discussed in text, but may have missed it.

The horizontal dissipation reduce when storms are removed (this is shown in pannel f). But pannel h represent the fraction of the total dissipation achieved by vertical processes. The increase of this fraction, suggests that the reduction of horizontal dissipation is larger than the reduction of vertical dissipation. This is discussed in Section 5.4 - "Horizontal resolution".

Figure 15

- the abbreviation VFORM should be included after "vector form" in the caption.

Thanks, this has been included, as well as "CEN2".

---

## Author Comment (AC1)

This manuscript addresses the fate of energy (mostly near-inertial energy) imparted by Southern Ocean storms and its effect on the Southern Ocean overturning circulation using a suite of semi-idealized high-resolution ocean models. The numerical experiments designed by the authors are suitable to address this important topic. Although the results on near-inertial energy generation, propagation and dissipation are somewhat expected, its effect on Southern Ocean overturning circulation is new. Furthermore, I haven't seen sensitivity studies of near-inertial energy budget to model numerics at such high resolutions before. Therefore I think this manuscript will make a useful and necessary contribution to this topic. I have some comments for the authors to consider, but most of my comments are minor in nature.

1. The manuscript talks about "energy imparted by mid-latitude storms". Well, it is not strictly correct. The synoptic winds associated with mid-latitude storms not only generate near-inertial waves via time-varying wind stress, but also contribute significantly to the time-mean wind stress at mid-latitudes via the nonlinear dependence of wind stress on the wind. For example, Zhai et al. (JPO, 2012) showed that when the synoptic winds are included in the stress calculation, the wind power input to the ocean general circulation can increase by as much as 70%.

*Indeed, our model setup does not allow to evaluate the response of the ocean to modification of the time-mean wind stress by the storms : by construction the 10 year average wind stress is the same in the storms and no-storms simulations. This was a deliberate choice so we were sure that the sensitivity of the solution was not due at first order to changes in the time-mean wind stress. One another side, our focus is not only on the near-inertial energy input (as you raised/questioned below, a substantial part of the storm energy input also feed the balanced circulation). For this reason, we choose to maintain the initial version of the title. Nevertheless we add the following sentence in the abstract in order to clarify the scope of our study : ¨The forcing strategy ensures that the time mean wind stress is the same between the different simulations so the effect of the storms on the mean wind stress and resulting impacts on the Southern Ocean dynamics are not considered in this study.¨*

2. Recently, Rath et al. (JGR, 2013) found that accounting for the ocean-surface velocity dependence of the wind stress leads to a large reduction of wind-induced near inertial energy of approximately 40% in their 1/10 degree Southern Ocean model. This relative wind damping effect seems to be a very important way of dissipating near inertial energy in the ocean. When you force your model directly with wind stress that knows nothing about the surface ocean currents, this relative wind damping effect is absent. I think this issue should at least be discussed, given that this manuscript is about energy dissipation.

*The following paragraph has been added to the discussion section (7.2 Energy pathways) : ¨Using a 1/10° model of the Southern Ocean, Rath et al. (2013) found that accounting for the ocean-surface velocity dependence of the wind-stress decreases the near inertial wind power input by about 20% but also damps the ML near-inertial motions leading to an overall ~40% decrease of the ML near inertial energy. Overall, this damping effect is found to be proportional to the inverse of the ocean-surface-mixed-layer depth. In our set of simulation, we do not include any wind stress dependence on ocean-surface velocity which remains a debated subject (Renault et al. 2016). Our main motivation for doing*

*so was to ensure that the mean wind stress remains the same between the different model experiments that have been performed in this study. Nevertheless, we should keep in mind that we miss a potentially important dissipative process for the NIWs.¨*

3. line 19-20 on page 3. Should be "2000 km long" and "3000 km wide".

*Corrected.*

4. line 26 on page 3 and in many other places in the manuscript. It should be "$3 \times 10^4$".

*It has been corrected here and elsewhere.*

5. The last sentence on page 3 needs to be rephrased. It is not clear to me what you want to say here.

*This sentence has been rephrased as follows : "Our horizontal resolution ≥ 1km and the hydrostatic approximation used to derive the model primitive equations do not permit the proper representation of upward radiation and breaking of internal lee waves (Nikurashin et al., 2011). Nevertheless, the deep flows impinging on bottom irregularities generate fine-scale shear which enhances dissipation and mixing close to the bottom, as generally observed in the Southern Ocean (Waterman et al. 2013). "*

6. In the model configuration section on page 4, there is no information about the existence of the background wind forcing without storms at all. It should not be left completely in the appendix.

*The description of the background windstress has been removed from the appendix and included in Section 2.1.*

7. line 21 on page 4. Brackets are needed for "2012".

*Corrected.*

8. page 6. (2) also includes diffusive energy transport in the vertical direction, not just energy dissipation?

*The energy transport by diffusion is removed by the vertical integration.*

9. line 12 on page 7. The average KE exceeds......, but the caption of Fig. 6a says "EKE"?

*This has been corrected in the text.*

10. line 1 on page 9. What is WKB stretched CW and CCW? Need to explain it or at least point the readers to the appendix.

*We now refer to appendix B.*

11. line 21-22 on page 9. Why does the storm lead to strengthening of the eastward current?

*This strengthening is due to the position of the mean eastward current which is not symmetric with respect to y=1500 km but intensified on the northern portion of the domain (as in Fig. 2 of Abernathey et al. 2011), so the domain average additional zonal wind work imparted by the storm is nonzero and positive. This is now mentioned in the manuscript in Section 4.2.*

12. line 25 on page 9. The Coriolis force is perpendicular to the current, and therefore should not do any work?

*Indeed, the contribution of the Coriolis force to energy budget should be zero. But errors due to the staggering of the Arakawa C grid turn its contribution to the kinetic energy balance nonzero (but very weak compared to the other term of energy equation). This is now mentioned in the manuscript (Section 4.2).*

13. line 17 on page 10. The authors should check what these papers say before citing them here.

*We retained the references to Garrett 2001, Blaker et al. 2012, Komori et al. 2008, but indeed, we found the references to Zhai et al. 2005 and Zhai et al. 2009 not adequate and we remove them. We also add a reference to Zhai et al. 2004 (GRL) on advective spreading of NIW that is the one we should have cited first.*

14. line 21 on page 10. You need to explain how the strength of the storm activity is varied seasonally here, or at least point the readers to the appendix.

*We add the following sentence : "The seasonality of the storms is included by seasonally varying the maximum wind stress of the storms from 0.75 N m$^{-2}$ in austral summer to 1.5 N m$^{-2}$ in austral winter (see details in Appendix A)."*

15. line 31 on page 10. Again, why does the mean KE increase in response to the storms?

*Again, this is because the mean ACC is not centered at y=1500km and not symmetric.*

16. line 7 on page 11. How do you define the mixed layer ?

*The mixed-layer depth is computed with a fixed threshold criterion (0.2°C) relative to the temperature at 10 meters. This is indicated in the caption of Figure 2.*

17. line 23 on page 13. The effective frequency should be "f+/2", according to Kunze (JPO, 1985)?

*We used the effective frequency as defined in Kunze et al. 1995 that was describing inertial oscillations from a reference frame rotating with a geostrophic flow with relative vorticity $\zeta$ . Since our reference*

*frame is only rotating with earth you are right : the correct definition of the effective frequency is $f+\zeta/2$ (Kunze, JPO, 1985). This has been corrected in the text and the figure has been updated.*

18. line 12 on page 17. Should be "easy comparison".

*This has been corrected.*

19. line 22 on page 17. I assume that changes in air-sea fluxes are due to the feedback term in surface heat flux forcing? If so, need to say it.

*Yes, the changes in air-sea fluxes are due to the feedback term. This is now mentioned in the manuscript as follows : "The change in the air-sea fluxes is due to the feedback term that act to restore model SST toward a SST climatology."*

20. lines 13-14 on page 19. It is not clear to me why you conclude that your results are noticeably different from the previous two studies, since your results also show that the majority of energy imparted by the storms is dissipated within the top 200 m.

*We agree that the overall conclusion is coherent with previous studies. This sentence has been rephrased as follows : "Our results are in good agreement with these studies: ..."*

21. line 18 on page 19. What is that part of wind work that is not near-inertial wind work? Wind energy input to the surface Ekman currents?

*The part of the wind work which is not near-inertial feed the subinertial and mean circulation.*

22.Figure 5. Three different color maps are used. Is this necessary? Why is 17 days after the passage of the storm is chosen? Is it to give enough time for the near-inertial waves to reach the base of the anticyclonic eddy?

*We choose three different colormaps to help distinguish the different fields : horizontal velocities, vertical velocities, shear. Yes we choose 17 days so it gives enough time to the waves to reach the base of the anticylonic eddy (as suggested by Figure 8b). This is now mentioned in the figure caption.*

23. Figure 8. a) and b) are plotted differently (one on log scale and one not), which makes it hard to compare them quantitatively. In the caption for c), the symbol "< >" appears messed up on my printed-out version.

*Figure 8a and 8b are now both plotted in log scale. The <> symbols have been corrected.*